# A biologically inspired repair mechanism for neuronal reconstructions with a focus on human dendrites

**Moritz Groden**[1]*, **Hannah M. Moessinger**[2], **Barbara Schaffran**[2,3], **Javier DeFelipe**[4,5], **Ruth Benavides-Piccione**[4,5‡], **Hermann Cuntz**[1,2,6‡], **Peter Jedlicka**[1,7‡]

**1** 3R Computer-Based Modelling, Faculty of Medicine, ICAR3R, Justus Liebig University Giessen, Giessen, Germany, **2** Ernst Strüngmann Institute (ESI) for Neuroscience in cooperation with the Max Planck Society, Frankfurt am Main, Germany, **3** Center for Neurodegenerative Diseases (DZNE), Bonn, Germany, **4** Laboratorio Cajal de Circuitos Corticales (CTB), Universidad Politécnica de Madrid, Spain, **5** Instituto Cajal (CSIC), Madrid, Spain, **6** Frankfurt Institute for Advanced Studies, Frankfurt am Main, Germany, **7** Institute of Clinical Neuroanatomy, Neuroscience Center, Goethe University, Frankfurt am Main, Germany

‡ These authors are joint senior authors on this work.
* moritz.groden@informatik.med.uni-giessen.de

**Data Availability Statement:** The fix_tree function and the fix_tree_UI as well as all the code needed to operate the repair algorithm is publically

## Abstract

Investigating and modelling the functionality of human neurons remains challenging due to the technical limitations, resulting in scarce and incomplete 3D anatomical reconstructions. Here we used a morphological modelling approach based on optimal wiring to repair the parts of a dendritic morphology that were lost due to incomplete tissue samples. In *Drosophila*, where dendritic regrowth has been studied experimentally using laser ablation, we found that modelling the regrowth reproduced a bimodal distribution between regeneration of cut branches and invasion by neighbouring branches. Interestingly, our repair model followed growth rules similar to those for the generation of a new dendritic tree. To generalise the repair algorithm from *Drosophila* to mammalian neurons, we artificially sectioned reconstructed dendrites from mouse and human hippocampal pyramidal cell morphologies, and showed that the regrown dendrites were morphologically similar to the original ones. Furthermore, we were able to restore their electrophysiological functionality, as evidenced by the recovery of their firing behaviour. Importantly, we show that such repairs also apply to other neuron types including hippocampal granule cells and cerebellar Purkinje cells. We then extrapolated the repair to incomplete human CA1 pyramidal neurons, where the anatomical boundaries of the particular brain areas innervated by the neurons in question were known. Interestingly, the repair of incomplete human dendrites helped to simulate the recently observed increased synaptic thresholds for dendritic NMDA spikes in human versus mouse dendrites. To make the repair tool available to the neuroscience community, we have developed an intuitive and simple graphical user interface (GUI), which is available in the *TREES toolbox* (www.treestoolbox.org).

available within the TREES toolbox at www.treestoolbox.org.

**Funding:** This work was supported by BMBF grants (No. 031L0229 – HUMANEUROMOD to P. J.; No. 01GQ1406—Bernstein Award 2013 to H. C.), DFG grants (CU 217/2-1; JE 528/10-1 467764793) and the von Behring Röntgen Foundation (to P.J.). The funders had no role in study design, data collection and analysis, decision to publish, or preparation of the manuscript.

**Competing interests:** The authors have declared that no competing interests exist.

## Author summary

Reconstructing neuronal dendrites by drawing their 3D branching structures in the computer has proved to be crucial for interpreting the flow of electrical signals and therefore the computations that dendrites perform on their inputs. These reconstructions are tedious and prone to disruptive limitations imposed by experimental procedures. In recent years, complementary computational procedures have emerged that reproduce the fine details of morphology in theoretical models. These models allow, for example, to populate large-scale neural networks and to study structure-function relationships. In this work we use a morphological model based on optimised wiring for signal conduction and material cost to repair faulty reconstructions. This is particularly relevant for human hippocampal dendrites, as data on their morphology is scarce and valuable but frequently compromised by technical limitations. Interestingly, we find that our synthetic repair mechanism reproduces the two distinct modes of repair observed in real dendrites: regeneration from the severed branch and invasion from neighbouring branches. Our model therefore provides both a useful tool for single-cell electrophysiological simulations and a useful theoretical concept for studying the biology of dendrite repair.

## Introduction

It is well established that dendritic geometry affects neuronal function [1–5]. For example, a change in dendritic size or topology may significantly alter the neuronal firing behaviour [6–9] in a possibly selective manner [10]. Several studies on the morphology and electrophysiology of human neurons have revealed their specific enhanced computational features [11–14]. However, systematic investigations of the relationship between the structure and the electrophysiological properties of human dendrites in computational models remain challenging [15, 16] since complete 3D reconstructions are scarce [17]. The sparse anatomical data that is available usually comes from both autopsies of healthy donors and biopsies of patients with brain diseases such as epilepsy or brain tumours [18–20]. These diseases may significantly alter the morphology and electrophysiology of a neuron [21, 22], resulting in severely impaired cognitive function [23]. Such pathological dendritic data may limit scientific conclusions if they are interpreted as coming from healthy controls.

Moreover, the reconstruction of labelled cells is often incomplete due to technical limitations related to staining processing, and intracellular injection requirements [24–26, see for example in Fig 1A–1C]. In addition, staining dyes injected into larger neurons, such as those in the human brain, often fail to reach the most distal dendritic regions [27, 28]. Such incomplete reconstructions, further limit the ability to study dendritic anatomy. However, the characterisation of morphological differences between human and other species' neural circuits [26, 29–31] is of great importance, since they have been shown to lead to distinct computational properties [11, 32–34]. Therefore, more complete human morphologies are urgently needed for a better understanding of human neuronal physiology and pathophysiology and for the creation of realistic computational models of human dendrites [5, 13, 15, 35].

Solutions to implement repair tools for morphological neuronal reconstructions have been proposed in the past [36]. These models have usually focused on detecting and fixing or removing artefacts that may occur during the reconstruction process, such as neurites that are not properly connected to the soma, removing segments of zero length, or adjusting dendrites that cross each other [37, *NeuroR*,]. **Ref**. [37] shows a morphological repair model, growing exclusively from severed branch ends. Electrophysiological analysis of such repairs, or of

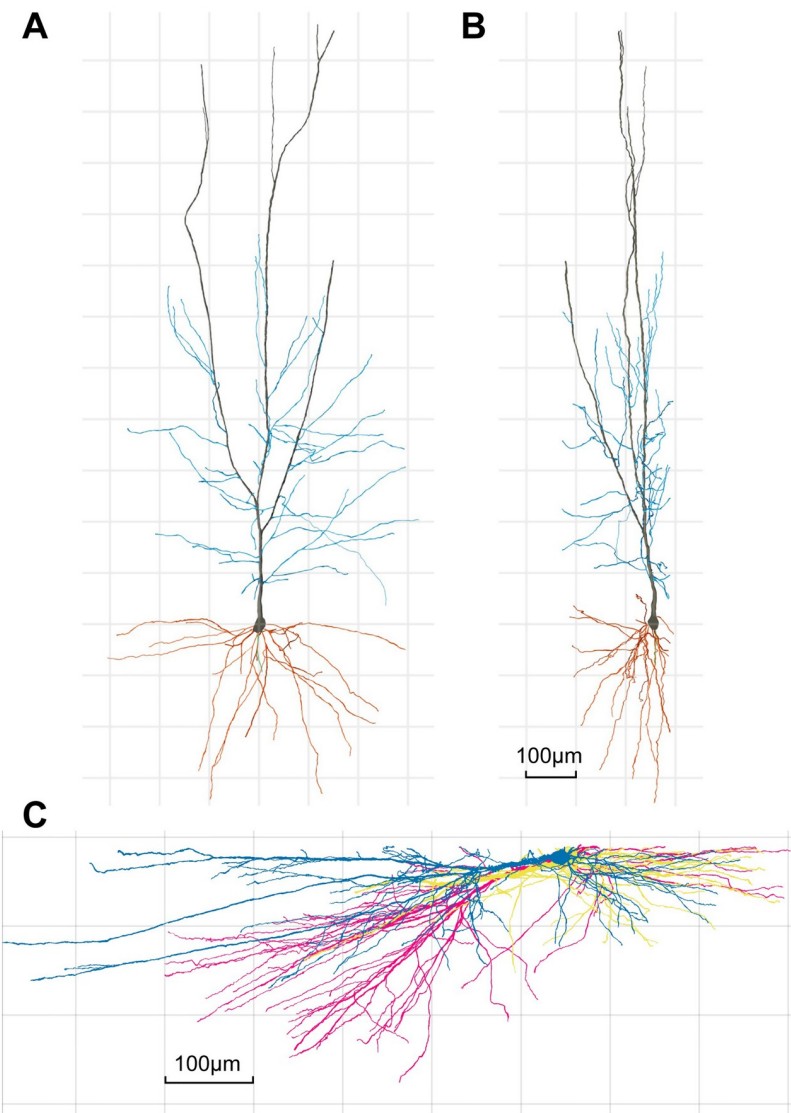

**Fig 1. Examples of human CA1 pyramidal reconstructions that were cut in the same plane during tissue sectioning.** Example of a 3D-reconstructed human CA1 pyramidal cell shown on the XY **A**, and YZ **B**, planes, to illustrate that, due to technical limitations, part of the dendritic arbour closest to the surface of the slice from which the cell soma is injected (typically at a depth of $\sim 30\mu m$ from the surface) is lost. Axon, main apical, collateral and basal dendrites are shown in green, black, blue and orange, respectively. Scale bar (in panel **B**) = $100\mu m$. **C**, Three human CA1 pyramidal neuron reconstructions (yellow, pink and blue) from the same preparation viewed from the side. Raw data from [26].

different neuron type repairs has not yet been conducted. Other morphological growth models are usually implemented as stochastic procedures based on branch probabilities and the number of branching events [38–40]. These branch probabilities are sampled from experimental distributions. This results in a large number of model parameters that must be adjusted to generate different neuron types. Adding entirely new branches to existing dendritic trees is not part of such tools.

For these reasons, in this work we investigated whether *in silico* dendritic growth algorithms based on optimal wiring [41–43] are able to complete incomplete morphology

reconstructions by adding missing parts of the dendritic tree that reproduce real structures. Optimal wiring principles allow the dendritic structure to be described by locally optimised graphs, in which total length and path length are minimised [44, 45]. An algorithm that weighs these two factors by a balancing factor *bf* can generate synthetic trees that reproduce biological dendrites [41, 46]. The impact of the balancing factor is showcased in Fig 2A by repairing an artificial 2D morphology using different values of *bf*. A small *bf* (close to or equal to 0) favours minimising of total cable length, as opposed to the direct path length to the soma (or the signal travel time to the soma). In turn, short path lengths are favoured when *bf* is large (close to or equal to 1). Once target points (target points are successively connected to the dendritic tree according to optimal wiring principles weighted by *bf*) are distributed within a cell-type specific dendritic density field, they can be connected to a tree structure according to these optimised wiring costs in *e.g.* fly [47] or mouse [48] dendrites as well as in some axons [49]. Given the general applicability of the method, here we investigate whether such morphological modelling can also be used to better understand and implement dendrite repair.

The biological system that inspired our regrowth algorithm was the nervous system of the *Drosophila* larva with so-called da (dendritic arbourisation) neurons [50]. These are divided into four classes based on their dendritic pattern, classes I–IV. Class IV da neurons grow predominantly in a two-dimensional space [51] and are well known to regrow their dendrites after dendriotomy [52, 53]. Almost 98% of all proximally lesioned dendrites showed regrowth, as measured by receptive field coverage after lesioning. Interestingly, in some cases the cut dendrite regenerated from the site of its lesion, and in others the field was covered by invading neighbouring branches of the same neuron, showing a bimodal distribution of dendrite regrowth [52].

In the work presented here, we report that our synthetic growth algorithm has the ability to mimic biological regrowth and to reproduce its two observed modes. In addition, regrowth can be tuned to emerge exclusively from the known incomplete ends of severed dendrite morphologies. Taking advantage of these features, we build a *TREES toolbox* function *fix_tree* and a user interface *fix_tree_UI* to complete dendritic reconstructions inspired by biological regrowth.

## Results

### A repair mechanism inspired from biology

To develop a repair algorithm for incomplete/damaged dendrites of nerve cells based on biologically inspired mechanisms, we first simulated and analysed the synthetic regrowth of dendrites characterised in a controlled experimental setting. Class IV da neurons of *Drosophila* larvae are a useful and well-studied experimental model system to investigate dendritic growth following dendriotomy [52–54]. To simulate the repair mechanism, seven reconstructions of class IV neurons [55] were taken from *NeuroMorpho.org* [56, 57]. The location where the cell was cut was chosen as a random branch point of the original morphology. The root of the severed branch (including the branch point) was used as a reference to determine the type of regrowth following the lesion. Branches growing back from this node during the repair process were defined as regenerated. Branches, that innervated the lesioned area but did not originate from the lesion node, were considered to be invading the space made available by the lesion.

We implemented a regrowth protocol, using newly distributed target points within the region of the severed branch, and replicated the stochastic regrowth *in silico* (see Methods). Regrowth based solely on optimal wiring principles, balancing path length and total wiring cost [41], successfully reproduced the main features of the dendrites. Importantly, our model

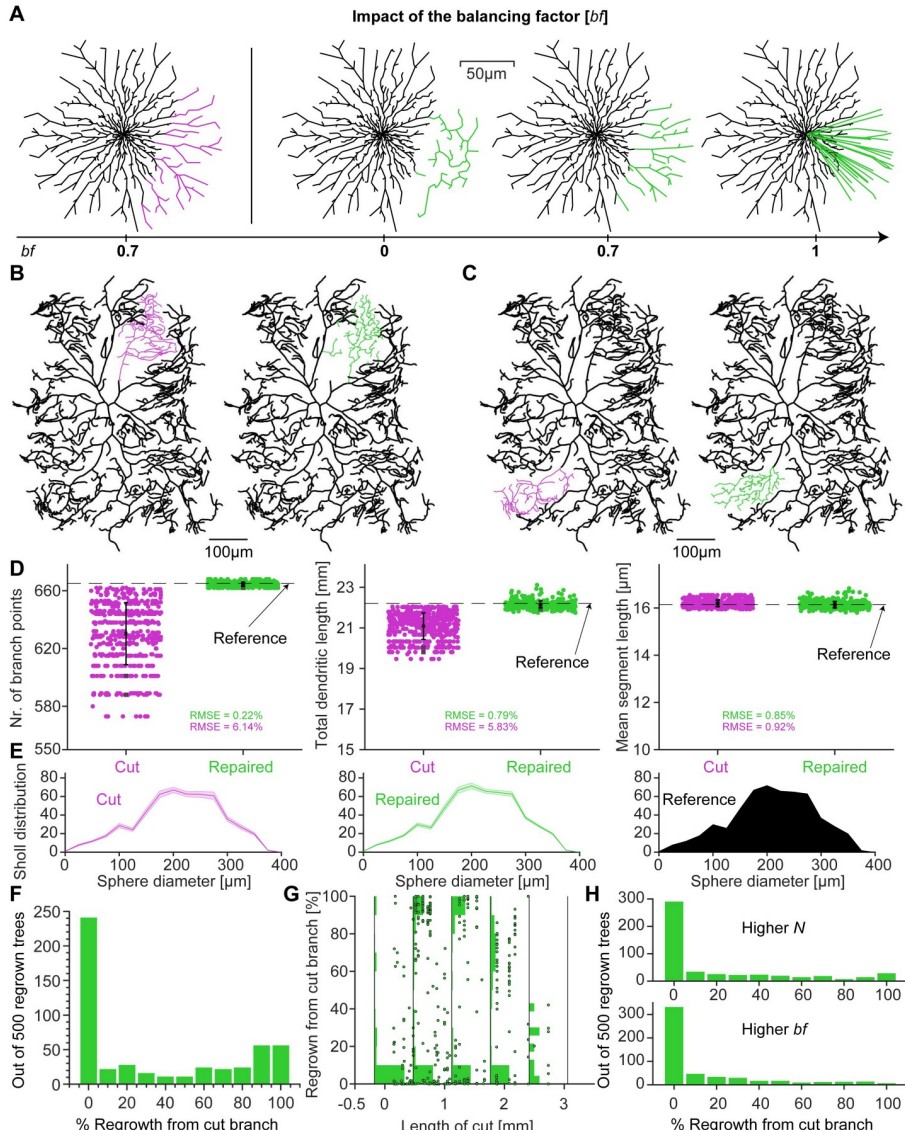

**Fig 2. Reproduction of biological regrowth of severed class IV *Drosophila* neurons. A**, The impact of the balancing factor *bf* demonstrated by repairing a 2D artificially created morphology (left with cut dendrites in magenta). Repairs with different balancing factors on the right (repaired dendrites in green). *bf* = 0 favours the minimisation of the overall cable length whereas *bf* = 1 optimises the direct path length towards the soma. **B** and **C**, *Left*, Reference *Drosophila* larva class IV morphology in which the branches that will be severed deliberately are marked in magenta. **B**, *Right*, Example of repaired dendrite where invasion has occurred from adjacent branches marked (green). **C**, *Right*, Sample repair where the severed branch regenerated from the cut end (green). **D**, Morphological statistics of the regrown dendrites from **B** and **C** (green) and 498 other random cuts. The repaired morphologies were compared to the original reference neuron (black + magenta in **B** and **C**) shown here as the black dashed line, as well as to the cut dendrites (magenta). The root mean square errors (RMSE) indicate the deviation from the reference value as a percentage of the reference value. The examples shown in **B** and **C** are represented by the darker square data points. **E**, Average Sholl distribution for the cut and repaired morphologies from **D** with standard deviation as shaded area. The reference distribution is shown in black. **F**, Histogram for 500 regrown dendrites using our repair function, showing the percentage of regenerated branches. **G**, Percentage of regenerated branches as a function of the size of the removed branch. The points show the actual data as an overlay to the binned normalised histograms of the data points. The bin boundaries are indicated by the black vertical lines. **H**, Histograms for 500 regrown morphologies as in **F** but for a higher number of target points *N* and a higher balancing factor *bf*.

replicated the experimentally observed bimodal distribution of branch regeneration versus invasion from neighbouring branches:

The model, just like the experiments showed two different possibilities for regrowth. Sometimes the synthetic regrowth invaded the available space, with new branches emerging from adjacent branches (Fig 2B, the branches in magenta on the left were intentionally severed and then repaired, as shown by the synthetic branches in green on the right). At other times, the repair algorithm showed complete regeneration of the lesioned area by a branch originating from the severed node, as shown in the example in Fig 2C with similar colours. These two modes of synthetic regrowth were in close agreement with similar observations in the experiment of **Ref**. [52], indicating that optimal wiring principles may be sufficient to explain these experimental observations (Fig 2B and 2C). The authors of **Ref**. [52] state that class IV da neurons regenerate in an all-or-none fashion, meaning that a severed branch will either regrow or fail to do so entirely. The latter will leave the vacant area open to invasion by neighbouring branches. They report that in about 50% of branch injuries the severed stem would regenerate successfully, resulting in a bimodal distribution.

Reconstructions from *NeuroMorpho.org* allowed us to generate synthetic cells that matched the branching statistics of class IV da neurons [42, 43, 58]. Morphologies, quantified by number of branches, total dendritic length and mean segment length (one segment is measured from one branch point to the next), were similar in synthetic cells grown using the minimum spanning tree (MST) algorithm [41] compared to reconstructions from biological cells (Fig 2D) which is underlined by the reported RMSEs as a percentage of the reference value. Furthermore the synthetic cells match the Sholl distribution of the reference well (Fig 2E). In summary, the algorithm captured the structure of the synthetic trees to such an extent that the morphology could be recovered after removing parts of the tree.

A summary of 500 different cuts and synthetic regrowths clearly shows the bimodal distribution between regeneration and invasion (Fig 2F). There was a distinct peak at 0% regeneration, *i.e.* 100% invasion, and a flatter distribution of larger percentages of regeneration in the case of the *Drosophila* larval class IV neurons. In ca. 50% of the cases the severed stem showed regeneration. There were no obvious relationships between the amount of invasion and model parameters or morphological features. When the results were dissected by the size of the severed branch in mm (Fig 2G), all types of regrowth were observed for all sizes of severed branches apart from when the severed branch was very large. In such a case invasion was more prominent. The bimodal distribution was seen for all other lengths of cut as indicated by the binned normalised histograms in Fig 2G. Only for very large and very small cuts did 100% regeneration become less likely. The exact amount of regeneration depended both on the density of new branches (higher $N$) and on the balancing factor *bf*, the trade-off in the optimal wiring algorithm between minimising the conduction time (*i.e.*, path length) and minimising the total cable length (Fig 2H). However, both regeneration and invasion were possible outcomes of the synthetic regrowth.

## Repair of different neuron types

We then tested whether our regrowth model could be used as a general tool, applicable to a variety of neuron types and different species including humans. Our previously established algorithmic generation of distinct dendritic trees of different neuron types depends on a single free parameter, the balancing factor *bf*, weighing material cost (*i.e.* cable length) against conduction time to the soma (*i.e.* path length) [41]. Based on recently established algorithms [59], our regrowth model is able to automatically estimate the biological *bf* from any incomplete (input) dendrite morphology. It also analyses the density profile of branch and termination

points based on the input neuron to be repaired, and distributes target points accordingly. The MST algorithm [41] then grows new branches along these target points, the number of which is set according to the density of branch and termination points of the input neuron and the size of the growth volume in which the target points are distributed. All parameters can also be adjusted manually. In this way, both highly branched (with low *bf*) as well as less branched morphologies with longer straight branches (with high *bf*) can be modelled.

The repair algorithm uses the `fix_tree` function which analyses the input tree to determine the growth parameters and sets up the target points in the growth volume. `fix_tree` then calls upon the `MST_tree` function to grow artificial dendrites. Examples of different cell type repairs are depicted in Fig 3, where panel **A** shows a mouse dentate granule cell [60]. This type of cell minimises predominantly the conduction time (path length) as compared to the material cost (cable length) with a high *bf*. The granule cell repair was able to accurately match the reference number of branch points, while the total dendritic length and mean segment length were less reliably reproduced but still significantly improved over the cut version (Fig 3B, see RMSE as a percentage of the reference value). The Sholl distribution (Fig 3C) of the repair also showed a significant improvement over the cut neuron. In this case, we used the conserved growth mode, which limits the regrowth process to the known cut branches. Interestingly, the execution of the procedure from Fig 2, where random branches were cut from the morphology and then repaired using the biological regrowth of the `fix_tree` function, revealed different distributions of regeneration and invasion in the different neuron types (Fig 3D). To demonstrate that the algorithm works for any neuron type, the same procedure was applied to a mouse Purkinje cell [61] with many branches on the right side of Fig 3. Purkinje cells are known to minimise the material cost more than the conduction time, exhibiting a low *bf*, Fig 3E–3H (same layout as for the granule cell). The morphological statistics were in good agreement with the reference, except for the mean segment length, which showed only a slight improvement as indicated by the RMSE. The histogram in Fig 3G shows the regeneration vs invasion statistics of the Purkinje neuron. Although still bimodal, regrowth from the severed branch appeared to be more likely in granule cells when compared to Purkinje cells and *Drosophila* larval class IV neurons (*c.f.* Fig 2). This may be due to the relatively high balancing factor in granule cells.

## Implementation of the regrowth algorithm in a new user interface

Next we used the regrowth algorithm, validated above using the dendrite regeneration data from *Drosophila* da neurons, to develop a new practical tool for repairing lesioned 3D-imaged and reconstructed dendrites. The model was then tested using a dataset of mouse CA1 pyramidal neurons provided by [26] (see more details in supplementary S1 Fig). The reconstructions of this dataset, like most others [27], are incomplete due to technical limitations (see above). We have generated an *in silico* model that utilises a graphical user interface (GUI) capable of fixing arbitrary neuron morphologies by adding synthetic dendritic branches to the existing incomplete reconstruction (Fig 4). The GUI allows the user to upload any 3D reconstruction and draw or upload any 3D or 2D region where dendrites are missing in the reconstruction. The algorithm then automatically grows the artificial dendrite into the specified volume, preserving the original morphology. This is done by distributing target points in the specified volume and successively connecting them to the input morphology (see Methods). As a reference for the anatomical tissue context, the user can upload a microscope image stack to serve as a background.

As demonstrated in Fig 4 *Top*, the image can highlight the different layers of the given brain region, *e.g.* the CA1 region of the hippocampus. This helps as an anatomical indication of

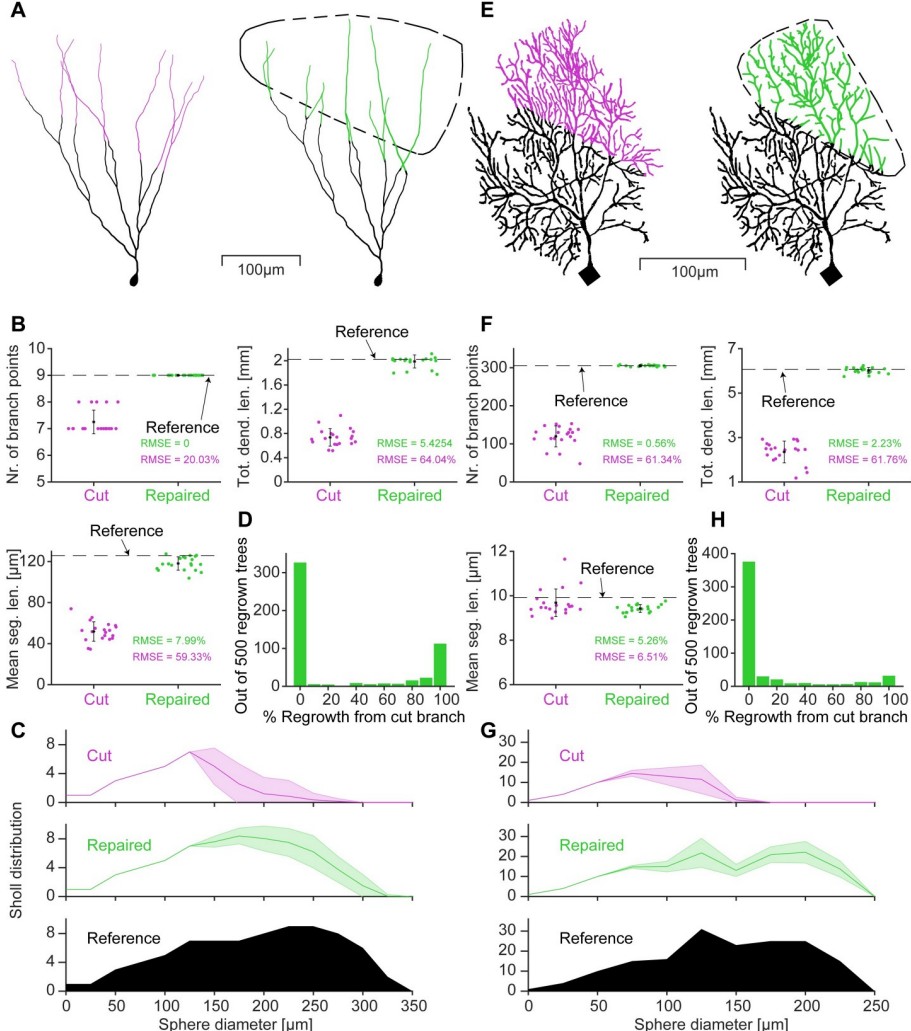

**Fig 3. Repair algorithm successfully restores removed dendrites of different neuron types with high, low and intermediate balancing factors. A** *Left*, Reference dentate granule cell morphology [Morphology from 60] with cut dendrites in magenta, *Right*, Repaired morphology with restored dendrites in green. The area enclosed by the dashed black line indicates the volume into which the dendrite has grown. **B**, Morphological statistics of 20 different cuts and repairs of the neuron shown in **A**. Total number of branches (*Top left*), Total dendritic length (*Top right*) and Mean segment length (*Bottom left*). The RMSE shows the deviation from the reference value as a percentage of the reference value. **C**, Average Sholl distribution for the cut and repaired morphologies from **B** with standard deviation as shaded area. The reference distribution is shown in black. **D**, Histogram of 500 regrown morphologies using our repair function `fix_tree`, with the percentage of the repair regrowing from the cut branch similar to Fig 2F. **E-H**, Same layout as in **A-D** but using a repaired mouse cerebellar Purkinje cell [Morphology from 61].

where the incomplete morphology might be repaired. The image can be set to the correct size and the morphology moved to the correct location using the image stack panel of the GUI. To draw a 3D target volume, the coordinates for its outline are selected with the cursor in at least two planes (*e.g.* x-y-plane and x-z-plane). Alternatively, the volume coordinates can simply be uploaded. Pressing the Repair button automatically estimates all parameters (see Methods) except the pruning parameters (truncation of terminal dendritic branches below a certain length threshold) and performs the repair. All parameters can also be adjusted manually by the user as well. As shown in Fig 4 *Bottom*, the GUI outputs the repaired morphology as well as

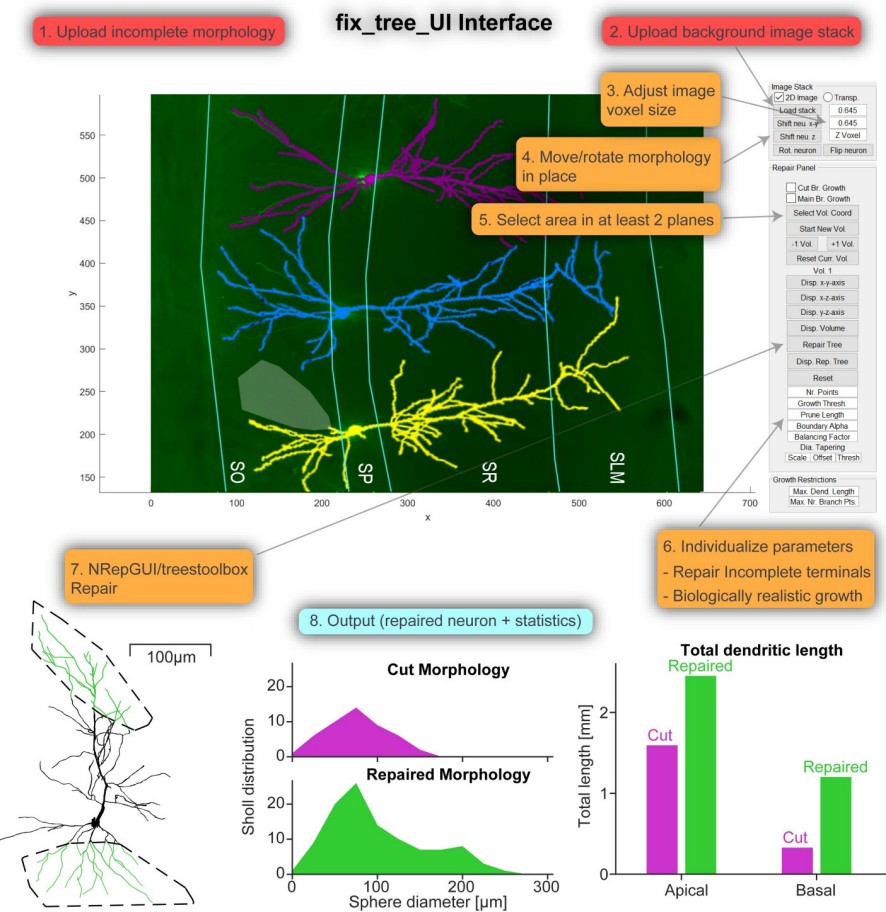

**Fig 4. A new software tool for the repair of dendrites with a graphical user interface (GUI).** *Top*, Example screenshot of *fix_tree_UI* (Neuron Repair Graphical User Interface). The numbers 1–8 represent the steps of successfully uploading a morphology and background image stack and repairing a missing region. *Bottom*, Showcase of the output of *fix_tree_UI* with the repaired neuron and two example statistics (the output contains more statistics than shown). Step one appears when launching the GUI asking the user to upload a morphology and step two can be initiated by clicking the "Load stack" button.

statistical morphological data comparing the input and output morphologies. If available, the user can also upload a reference morphology to be used as a template. The algorithm then matches the statistics of the repair to the reference reconstruction. In this way, the repair mechanism can be tested on sample data before being applied to data from actual incomplete reconstructions.

## Repair of artificially sectioned mouse CA1 pyramidal neurons

Next we tested our repair algorithm on mouse CA1 pyramidal neuron morphologies [26] (Fig 5A). To assess the quality of our repair algorithm, existing reconstructions were arbitrarily cut at different points and angles in the apical and basal arbour. The original morphology served as a reference and ground truth. The comparison between the reference and the repaired morphology showed the accuracy of the repair (Fig 5A, *Top*). Dendritic branching profiles [62] as a function of the distance from soma showed that the repair algorithm was able to restore the original dendritic shape (Fig 5). The Sholl profiles showed a significant improvement over the

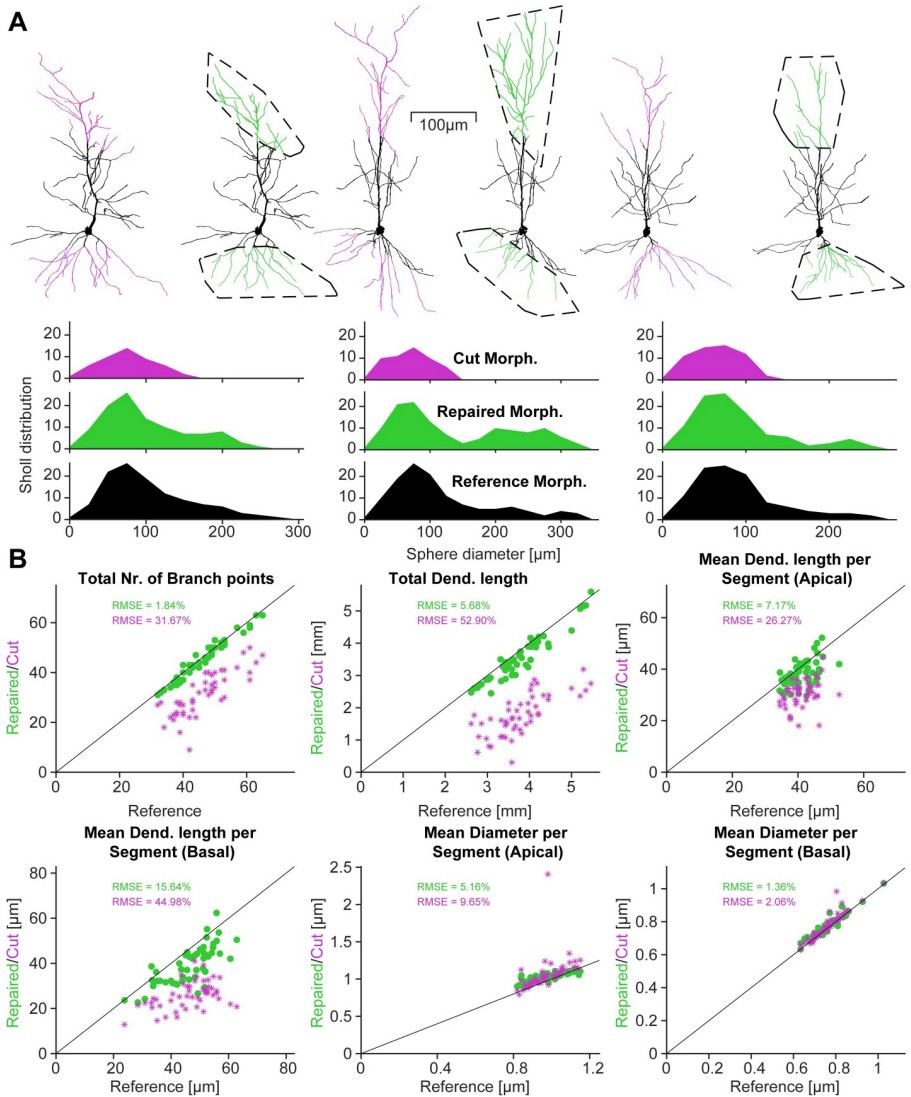

**Fig 5. The repair algorithm successfully recovered artificially removed dendrites from mouse CA1 pyramidal cells and restored their Sholl profiles. A**, Three example repairs of apical and basal dendrites of mouse CA1 pyramidal neurons [Reconstructions from 26]. For each repair, the left morphology is the reconstructed reference with cut branches in magenta and the right morphology is the repaired tree with regrown branches in green. The area enclosed by the dashed black line represents the 3D volume into which the artificial dendrites grew, corresponding to the convex hull of the severed dendrites (see Methods). The graphs below each repair show the distributions of Sholl intersections for the Cut, Repaired and Reference morphologies. **B**, Each graph shows the value of the repaired morphology (green dots) plotted against the value of the original morphology in black on the identity line. For comparison, the data points in magenta show the values for the cut morphologies. *Top, left to right* Total number of branch points, Total dendritic length, Average dendritic length per segment in the apical dendrite (one segment is measured from one branch point to the next). *Bottom, left to right* Average dendritic length per segment in the basal dendrite, Average diameter per segment in the apical dendrite, Average diameter per segment in the basal dendrite. The root mean square errors (RMSE) show the deviation from the reference values as a percentage of the reference value.

cut dendrites, but did not exactly replicate the reference. An exact match was not possible due to the stochastic nature of the repair mechanism. For example, in Fig 5A, (*Middle*) the two right side peaks in the Sholl profile were present in the repair, but were more prominent compared to the reference. As the dendrites were intentionally cut, the exact cut-off points were

known and the algorithm allowed new dendrites to grow exclusively from these incomplete branches (a forced conserved growth mode).

This additional growth mode was inspired by the regeneration observed in biology but was implemented here as a useful option in our software tool. The other mode allows the algorithm to grow new dendrites from any point of the existing morphology, preferably points that are close to the volume chosen for growth (invasion and regeneration). To repair incomplete morphologies we used the conserved growth mode, when the incomplete branches were known, such as in Fig 5A. This method allows the user to restore a part of the dendrite that they know should be there but could not be reconstructed from their tissue slice. The growth parameters for the algorithm are determined by analysing the remaining part of a cut dendrite, making repairs more difficult when only minimal dendritic material remains. Additionally, since pyramidal neuron main apical dendrites can branch, as observed in **Refs**. [26] and [31] there is a main growth option for the conserved growth mode (see Methods). With this option, a prominent straight main apical dendrite is grown first and then oblique dendrites are added. As the main apical dendrite is incomplete in all cases shown in Fig 5A, this option was used for the apical repairs. The extent to which the dendrites grow in a particular direction is given by the growth volume.

From a morphological point of view it is important to accurately analyse the shape and appearance of the neuron as well as the statistics of its morphology. Therefore, Fig 5B shows further details for the fine-grained morphological statistics of the pyramidal neurons from Fig 5A plus 47 additional mouse pyramidal neurons from [26] that have been cut and repaired in a similar fashion. The algorithm tries to fit the repairs to exactly match the number of branch points of the reference morphology (Fig 5B, *Top left*). The model also fits the total dendritic length well in most cases as shown in Fig 5B, *Top middle*. The remaining four statistics are the dendritic length per segment and the diameter per segment for apical and basal arbours (a segment is measured from one branch point to the next or from a branch point to a termination point). These results show that our model was able to reliably match the morphological properties of mouse CA1 pyramidal neurons in terms of shape and appearance as well as their statistical properties. The statistical agreement is also indicated by the RMSE values, which were approximately one order of magnitude smaller for each measure except for the mean dendritic length per segment in the basal dendrite and for the mean diameter per segment in the basal dendrite. For the latter, even the values for the cut dendrites were a close match. The improvement of the repair was measurable but not as significant. For the former, the repair approximately halved the RMSE.

## Repair of human CA1 pyramidal cell reconstructions

We also tested our method on incomplete human CA1 pyramidal neurons. We applied the repair algorithm to a dataset of CA1 pyramidal neurons from **Ref**. [26] depicted in Fig 6. Similar to the validation process carried out with the mouse reconstructions (Fig 5), we first applied our repair algorithm to the original reconstructions from Fig 6D. In particular, the basal dendritic arbour and the most distal apical dendritic collaterals and tufts were reconstructed. The results of these extensions are depicted in Fig 6E. The dendritic spanning fields of these artificially repaired morphologies are based on the layer limitation boundaries marked out in the slice image. Furthermore, it was assumed that CA1 pyramidal cell dendrites would extend more than halfway into the SLM when the soma of the neuron is close to the SP-SR boundary, in order to make synaptic connections with axons from the perforant pathway [31, 63]. S2 Fig shows examples of regions where dendrites were missing from the neuron reconstructions. The authors of **Ref**. [26] knew that human CA1 pyramidal neuron dendrites were present in

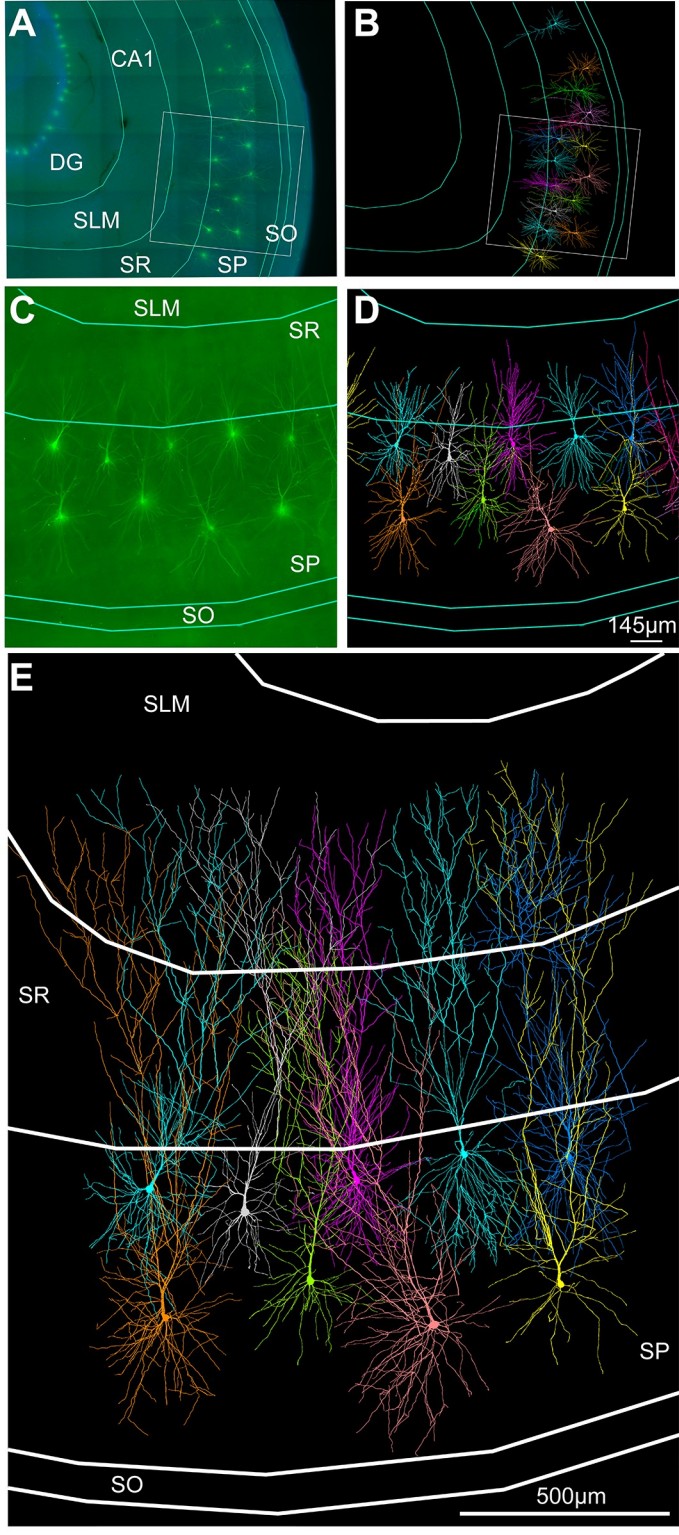

**Fig 6. Growth algorithm extends incomplete human CA1 pyramidal cell morphologies. A**, Confocal microscope image of the human hippocampal CA1 region (DG: dentate gyrus; SLM: stratum lacunosum moleculare; SR: stratum radiatum; SP: stratum pyramidale; SO: stratum oriens) with stained pyramidal cells and ROI (region of interest). **B**, Morphology reconstructions. **C**, ROI enlarged from **A**. **D**, ROI with overlays of originally reconstructed pyramidal cell morphologies by **Ref**. [26], which are incomplete due to experimental limitations (see text). Scale bar = 460 $\mu$m in **A, B**.

**E**, ROI showing morphologies from **D** that have been artificially extended in the apical and basal arbour, showing plausible completion of incomplete dendrites based on known layer-specific target growth regions. Each individual neuron has been given a different colour to distinguish the morphologies.

these regions, but could not reconstruct them because they were not visible in the microscopic images due to technical limitations and accidental lesions. We cannot evaluate the accuracy of the repair algorithm regarding the extended human neuron morphologies, as the model provides a prediction in this case.

## Restoration of firing behaviour in repaired mouse morphologies and predictions for human data

We tested whether our repair algorithm was able to restore the simple firing behaviour of the original morphology after regrowth of cut branches (Fig 7). We used a biophysical model from **Ref**. [64], implemented in mouse (Fig 7A) and human (Fig 7B) neuron morphologies. Somatic current clamp simulations were performed with the stimulation current increasing in five steps (from 0.16nA–0.24nA in Fig 7A and 0.26nA–0.46nA in Fig 7B) and lasting 500ms each. The cut neurons clearly displayed hyperexcitable firing behaviour (Fig 7). In the repaired neuron, the firing behaviour was restored as demonstrated by the F-I curves (Fig 7A and 7B insets). The firing rate of the cut neuron (magenta) was much higher than that of the reference (black) and repaired (green) neurons in both cases. The firing behaviour of the repaired neuron was close to that of the reference, although the firing rate was slightly lower. The discrete Fréchet distances between the cut-reference and repaired-reference electrophysiological curves show that the improvement of the repair was substantial (Mouse: cut-reference = 49.11; repaired-reference = 6.63; Human: cut-reference = 71.44; repaired-reference = 2.00). The Fréchet distance was approximately one order of magnitude smaller for the repair in both cases. To study the effect of different cuts and repairs on the electrophysiological behaviour we performed the same simulations as in Fig 7A and 7B but for different cuts on the same morphologies. The results are presented in S3 Fig, where the reference F-I curve is plotted in black next to the average of 20 different repaired lesions in green with the standard deviation (cut morphologies in magenta). There was no notable difference in the recovery of firing behaviour for the different cuts, which is supported by the discrete Fréchet distances between the curves of the average repaired, reference and cut morphologies (Mouse: cut-reference = 55.32; repaired-reference = 6.92; Human: cut-reference = 24.28; repaired-reference = 3.15). The Fréchet distances for the average of the different repairs were similar to those shown in Fig 7A and 7B. We conclude that using our repair tool to restore lost dendritic material can lead to the recovery of the original neuronal excitability, when only incomplete data is available. Fig 7C shows an incomplete human CA1 pyramidal neuron that has been artificially extended (*c.f.* Fig 6). The extended version is closer to the actual size of the neuron before the reconstruction process. Consequently, the electrophysiological behaviour predicted for the extended morphology by the [64] model differs from the incomplete reference morphology, as excitability is reduced in the extended version (Fig 7C *right*). The F-I curve inset underlines the reduction in excitability, predicting an average gain of $-16.67Hz$ for an extension depicted in the figure.

To analyse the effect of repair on more detailed electrophysiological properties of a neuron, such as sag current, inter-spike interval (ISI) and adaptation index, we used a different model of CA1 pyramidal neurons from **Refs**. [65, 66]. The model in **Ref**. [64] does not include HCN channels or slow $K^+$-channels, which would produce a sag current or spike adaptation, respectively. Therefore, such measurements can only be applied when using the model in **Refs**. [65,

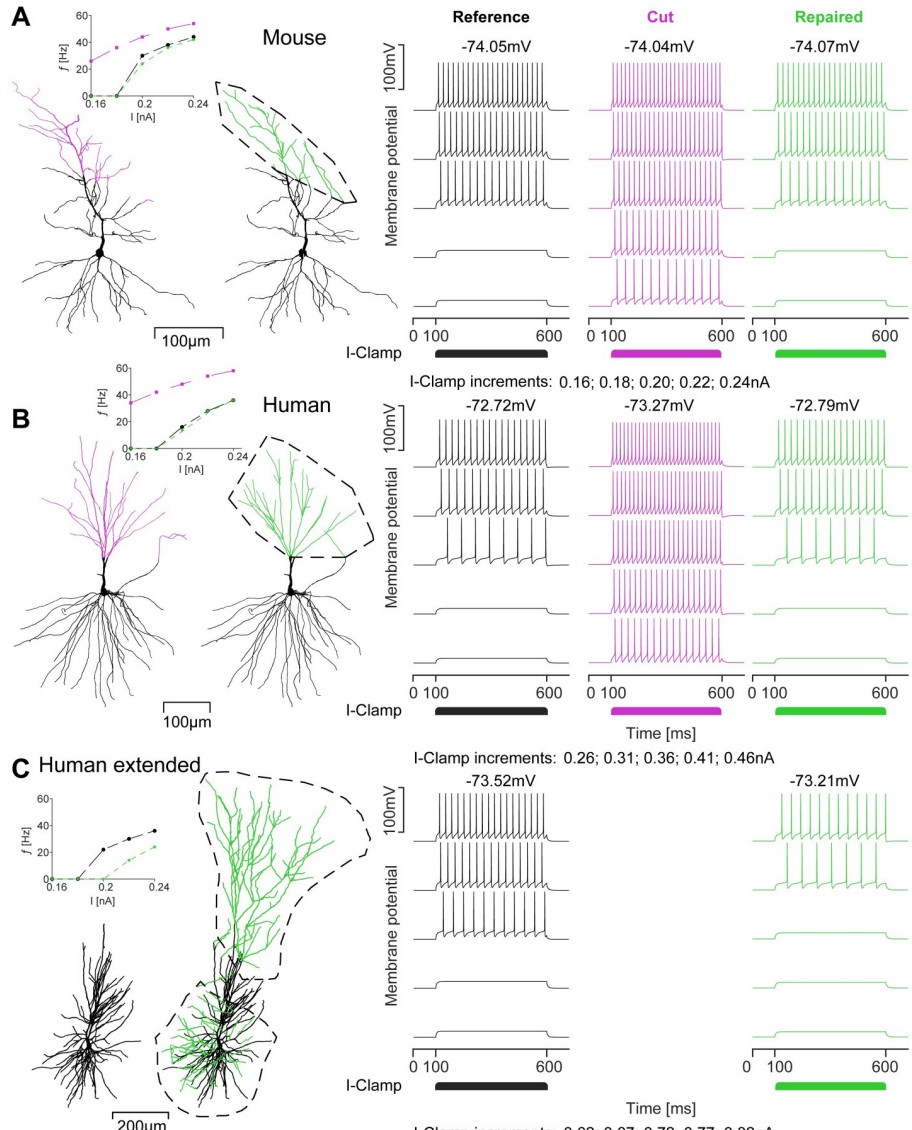

**Fig 7. The repair algorithm restores the electrophysiological behaviour of cut and repaired mouse pyramidal cells and allows for better predictions of neuronal function in human neurons. A**, CA1 pyramidal cell of the mouse. *Left*, Reference morphology, *Middle*, Repaired morphology with growth volume indicated by the black dashed line. Cut dendritic sections in magenta, repaired dendritic sections in green. *Right*, Somatic voltage traces induced by current injections in the soma of reference (black), cut (magenta) and repaired (green) morphology with resting membrane potentials (*Top*) and current clamp increments (*Bottom*). The inset on the left shows the F-I curves of the data on the right (same colour scheme). **B**, Human CA1 pyramidal cell. Same arrangement as in **A**. Repair restores the electrophysiological behaviour of the reference neuron. **C**, Prediction of the electrophysiological behaviour of an extended human CA1 pyramidal cell. Arrangement as in **A** but the reference morphology on the left is the full but incomplete reconstruction as provided in **Ref**. [26], which has been extended using the repair algorithm (*c.f.* Fig 6).

66], which was converted to T2N [60] in **Ref**. [67] such that the morphology can be exchanged. Here, we used a mouse CA1 pyramidal neuron morphology from **Ref**. [68] as was previously done in **Ref**. [69]. The morphology was cut and subsequently repaired using our algorithm in Fig 8. The model in **Refs**. [65, 66] became unstable for very large neurons such as those found in human tissue. The ion channel distribution in this model depends on the layers in CA1.

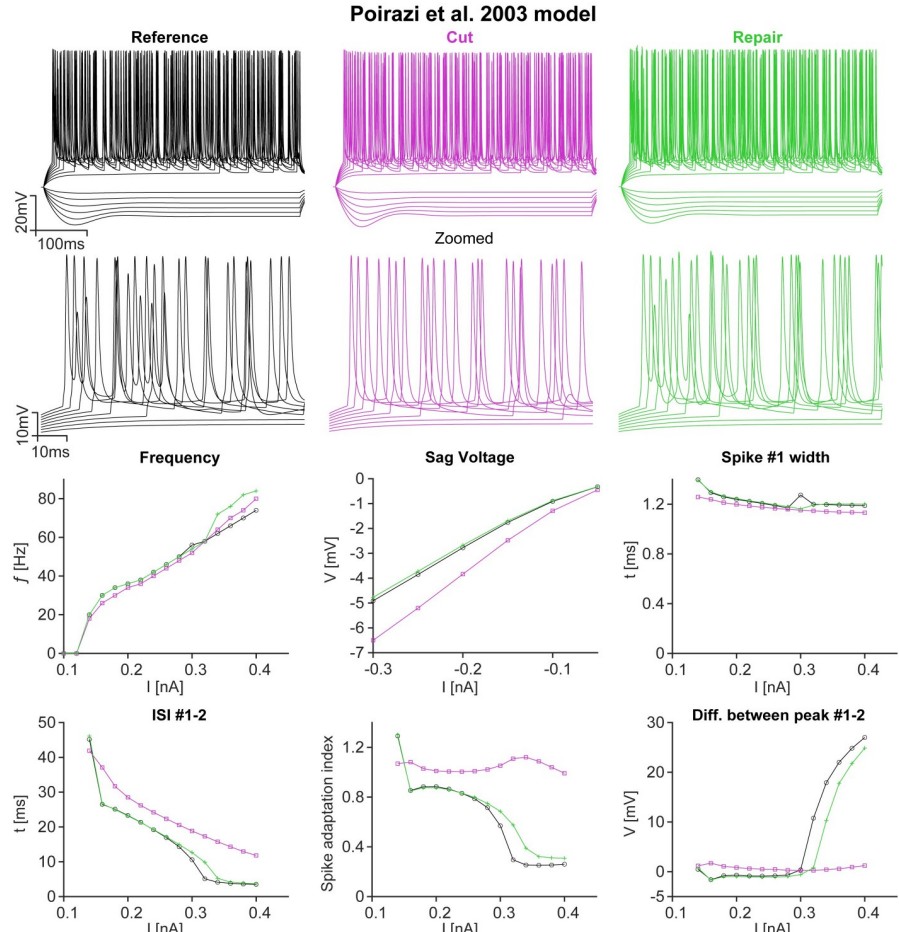

**Fig 8. Repairs restore the detailed electrophysiological behaviour of neurons.** *Top*, Example voltage traces of intentionally cut and subsequently repaired mouse CA1 pyramidal neuron current clamp simulations using the model by [65, 66] (neuron data by [68]). Reference in black, cut in magenta, repair in green (same colour scheme for the entire figure). A zoomed in version for the spiking traces is shown below for clearer visibility (zoom shows the beginning of the traces where spiking commences). *Middle (left to right)*, Firing frequency, sag voltage and half width of the first spike plotted against stimulation current increments. *Bottom (left to right)*, Interspike interval between the first and second spike, spike adaptation index and difference between the peak of the first and second spike plotted against stimulation current increments.

Laminar regions were well defined within the model for mouse neuron morphologies but not for human neurons. For this reason, human neurons have not been included in Fig 8. The top of Fig 8 depicts example voltage traces from current clamp simulations of a repaired mouse CA1 pyramidal neuron for the reference (black), the cut (magenta) and the repair (green) using the model in **Refs**. [65, 66] (a zoomed version is also depicted). The six plots below show different electrophysiological metrics for the three morphologies using the same colour scheme. In each case in Fig 8, we found that the repaired model (green line) was much closer to the reference (black line) than the cut model (magenta line), suggesting that repairing the morphology restored the electrophysiological behaviour of the neuron. The cut neuron exhibited a reduced firing frequency but a larger negative voltage sag. For small stimuli, the firing frequency of the repaired neuron was almost perfectly matched to the reference, but began to deviate for larger stimuli. The inter spike interval (ISI) between the first and second spike as

well as the spike adaptation index were all significantly increased for the cut neuron, with the half-width of the first spike being similar for all morphologies. Finally, the difference between the peak of the first and the second spike in the cut neuron was increased for small stimuli. For larger stimuli, the reference and repaired morphologies showed a steep increase, but the cut morphology did not. The spike adaptation index was larger for low stimuli in the reference and the repaired morphology, but decreased significantly for higher stimuli. In contrast, the adaptation index remained relatively constant for the cut morphology.

It has recently been reported that mouse dendrites in cortical pyramidal neurons have lower synaptic thresholds for NMDA spike generation than human dendrites [14]. To further demonstrate the restorative effects of our repair algorithm on the electrophysiological behaviour of rodent and human dendrites, we performed a computational analysis of their dendritic NMDA spiking. In particular, we were interested in the behaviour of incomplete morphologies that were extended beyond the reconstructed dendritic material (*c.f.* Figs 6 and 7). In Fig 9A, three morphologies were synaptically stimulated in their basal dendrites (highlighted colours; other dendrites in grey) at different Euclidean distances from the soma. The distances were scaled according to the size of the neurons, as the human morphologies were much larger than the mouse morphologies, defined by the percentage of the maximum possible distance within the basal dendrite. Using a passive version of the compartmental model of [64], AMPA and NMDA synapses were stimulated. The intensity of the stimulation was determined by the number of synapses distributed over sections of $20\mu m$.

Fig 9B shows example dendritic spike traces with increasing numbers of synapses, recorded at the site of stimulation, at 85.19% of the maximum possible distance from the soma. We compared a mouse pyramidal cell morphology with an incomplete human reference morphology and an elongated (extended) human neuronal morphology. We measured the peak voltage of NMDA spikes evoked by different numbers of synapses at different distances from the soma (Fig 9C). For each distance, 10 different dendritic locations at that specific distance were tested, as we found a lot of variation in the response (transparent dashed coloured lines) especially close to the soma (Fig 9C left). The mouse average peak voltage (solid blue) was generally the largest and had the steepest slope, whereas the voltage peaks in human (solid black) and human extended morphologies (solid green) were similar close to the soma. This is consistent with the findings of **Ref**. [14], who reported a lower threshold for eliciting NMDA spikes in mouse compared to human layer 2/3 pyramidal neurons. As one moved away from the soma, the response variation decreased in all morphologies with the peak of dendritic spikes in the human reference morphology (black) being more similar to the mouse neuronal morphology (blue), whereas the peak of dendritic spikes in the extended human neuronal morphology (green) was reduced. Thus, only the repaired human neuronal morphology maintained a higher synaptic threshold for NMDA spikes compared to its mouse counterpart. Therefore, an incomplete human neuron (incomplete due to reconstruction limitations) did not exhibit a different NMDA threshold on the outer parts of the dendrites. Completing the neuron with an extension using our algorithm restored this behaviour, increasing the threshold and therefore resulting in a lower peak voltage (Fig 9C *Right*). Thus the extended human neuron reproduced the findings of **Ref**. [14] more accurately than the incomplete human reconstruction. Since the human extended neurons (green) were even larger than the incomplete human neurons (black), the absolute distance of the stimulation site from the soma was not the same in these two cases. To investigate whether this discrepancy in distance had a significant impact on the analysis presented in Fig 9, we re-ran the same simulation as in Fig 9, where the absolute stimulation distances from the soma were the same in both human and human extended neurons (S4 Fig). The results in S4 Fig suggest that the difference in stimulation distance between the two human neurons had no significant effect on the NMDA spiking behaviour, as the results

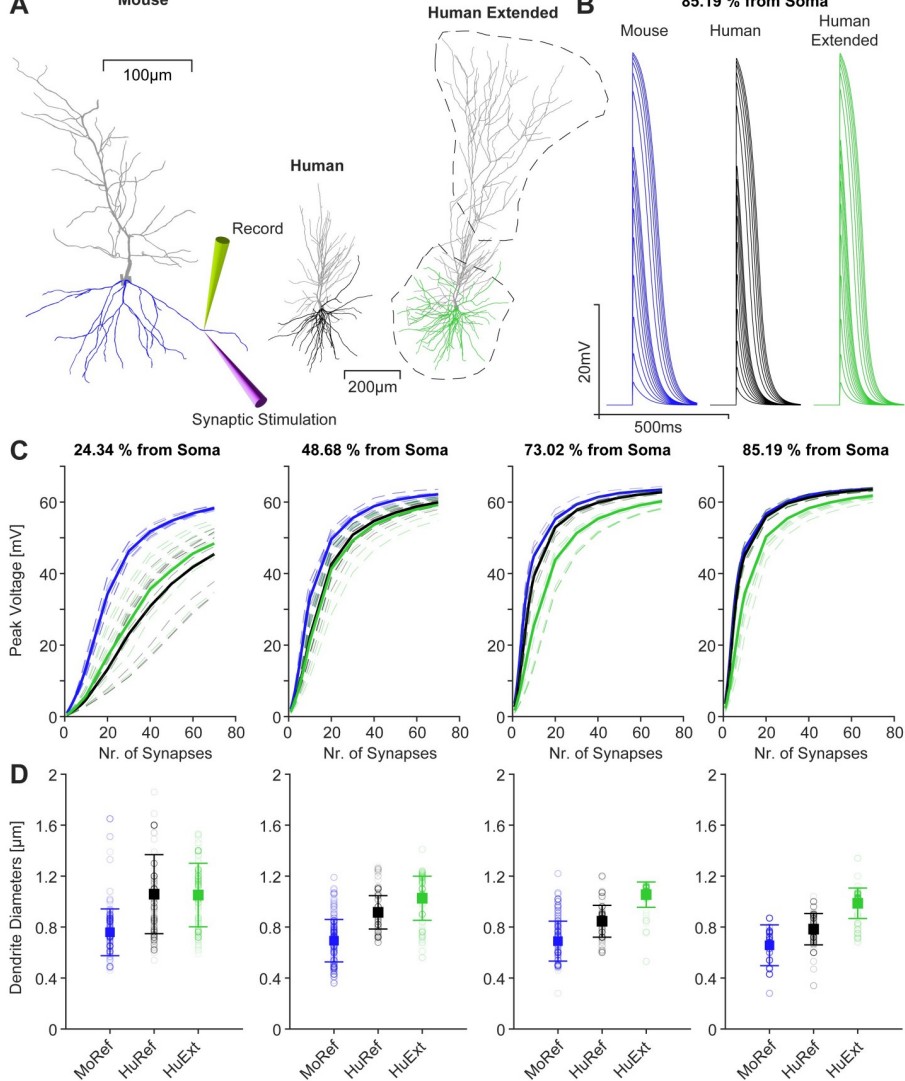

**Fig 9. Repairing neuronal dendrites is likely to improve simulations of NMDA spikes, which are reduced in extended human neurons compared to mice. A**, Mouse CA1 pyramidal cell with basal dendrites in blue. Stimulation and recording sites are indicated on the basal dendrite. *Right*, Human and human extended morphology with basal dendrites in green and black with the growth volumes indicated by the black dashed lines. The human extended version was created by extending the human reconstuction in **A**, *Middle*. **(continued) B**, Example dendritic NMDA spikes for a mouse (blue), human (black) and human extended (green) morphology at 85.19% of the maximum possible Euclidean distance in the basal tree away from the soma for each morphology, respectively. **C**, Peak NMDA spike voltage measured for different numbers of synapses at different distances from the soma in the basal dendrite, given as a percentage of the maximum possible distance in the basal tree (colour scheme as in **B**). For each distance, 10 different locations at that distance were tested (transparent dashed coloured lines). The average is shown as a solid line. The synapses were distributed over 20$\mu$m sections. **D**, Dendritic diameters for the locations described in **C**, with mean and standard deviation.

were similar to Fig 9. As the difference in stimulation distance in Fig 9 was only $\sim 10$–$20\mu m$, which was in the range of the size of the sections in which synapses were distributed, no significant effect was to be expected. Overall, in agreement with previous findings [14], the differences in NMDA spiking were associated with differences in dendritic diameter (Fig 9D). Close to the soma, dendritic diameter varied more than at distal locations, resulting in the large

variation in NMDA spikes close to the soma. At distal locations, diameters were consistently larger in the extended human neuronal morphology (Fig 9 green) than in the incomplete human neuronal morphology (Fig 9 black). With the reduced diameters, the incomplete human neuronal morphology (Fig 9 black) showed similar dendritic spikes to the mouse (Fig 9 blue). The differences in diameter were also found by Testa-Silva et al. [14], who provide evidence that the higher NMDA spike threshold in human neurons is likely due to larger dendritic diameters. Therefore, completing a human neuronal morphology by extending its dendrites using our repair tool may lead to more realistic simulations of NMDA spikes in human neurons.

## Discussion

In this work, we developed a morphological modelling algorithm based on optimal wiring to regrow previously severed dendritic branches. We report four main results. First, we show that the algorithm reproduces an experimentally observed bimodal distribution of dendritic regrowth, consisting of regeneration from lesioned branches and invasion from adjacent branches (Fig 2). Second, when applied to simulated lesions resulting in incomplete 3D morphologies, the repaired dendrites were morphologically similar to the original ones in terms of branching statistics and electrophysiological behaviour (Figs 5 and 7). Third, when applied to incompletely reconstructed human CA1 pyramidal neurons, the repair algorithm was able to improve their dendritic structure based on the known anatomical layer-related context (Fig 6). Finally, simulations of species-specific differences in NMDA spiking suggest that our approach may improve predictions of dendritic electrophysiology in incomplete reconstructions (Fig 9).

### Bimodal dendrite regrowth based on the trade-off between optimal cable length and conduction speed

The adapted *TREES toolbox* algorithm [41], which balances material cost, *i.e.* cable length of the dendrite, and conduction time, *i.e.* path length to the root [41], was able to successfully regrow dendrites of class IV da neurons of *Drosophila* after removing a part of the tree. The regrown dendrites were statistically similar to the cells under experimental conditions. Therefore, the same balancing factor (which quantifies the trade-off between cable length and conduction speed) underlying the same optimisation algorithm accounts for both a newly generated dendritic tree as well as for the completion of an already existing tree.

Intriguingly, both the computer model and the biological system [52] displayed a binary distribution of invasion vs. regeneration (Fig 2F). **Ref**. [52] investigated the regenerative capacity of class IV da neurons. Regeneration of class IV dendrites was a commonly observed phenomenon, with 49.4% showing regrowth from the lesioned stem in **Ref**. [52] [see also 53]. In cells where the severed stem did not regrow, neighbouring branches invaded the area and re-established coverage of the epithelial area by the dendritic network [52]. This modal response was clearly seen in both **Ref**. [52] as well as in our model. In general **Ref**. [53] reported a re-coverage of the lesioned area in almost all cases. In the case of 100% invasion, **Ref**. [52] reported retraction or stalling of the lesioned dendrite. It has also been observed in **Ref**. [53] that if they left a longer stump, the regeneration tended to initiate from there. Therefore it should be investigated how the site of dendriotomy influences invasion versus regeneration. Nevertheless, the close correspondence between the morphological statistics of the original and regenerated nerve cells clearly shows that a dendritic arbour has similar properties before and after the lesion, regardless of whether the empty space is invaded by non-lesioned branches or regenerated from the lesioned stem.

## Human and mammalian dendrite repair

Detailed anatomical data on human neurons remain limited [17]. For example, one of the largest public databases of neuronal morphologies, *NeuroMorpho.Org* [56, 57], contains human cell data in only $\sim 4.4\%$ of its entries. Neuroscientists face technical and ethical limitations that limit the acquisition of large datasets from the human brain [70–72]. However, there are structural and functional properties that are specific to the human brain and its neurons [73–80], which is why animal neurons cannot completely replace human ones [81]. Human neurons are not only larger but also more complex than those of for example macaques and marmosets [11]. Similar observations have been made when comparing humans and chimpanzees [82]. **Ref**. [83] found a wide range of differences between homologous mouse and human neuron types including gene expression, morphology and laminar distribution. To enable more complex brain functions, human neurons have probably evolved special mechanisms such as very strong excitatory synapses, which allow excitatory principal cells to trigger firing in local inhibitory interneurons via a single action potential [84]. Recent somatic and dendritic recordings in human neurons and their analyses have also revealed other human-specific electrophysiological properties [14, 30, 34, 35, 85–90]. These species-specific differences may contribute to the unique cognitive abilities of the human brain. Using our approach, such differences could be investigated by first predicting the shape and topology of the putative full morphology reconstruction (see Fig 6). In a second step the electrophysiological behaviour and how it differs from the reference reconstruction can be predicted by implementing compartmental models (see Fig 7). Extended reconstructions of full human neuronal morphology could also help to build more accurate human compartmental models, which can be done for any neuron type as the *fix_tree* function developed in our work is generally applicable.

Understanding the specific functionality of human neurons requires anatomically complete and reliable datasets of 3D human neuron reconstructions. Our repair tool could help address these issues and alleviate some of the difficulties.

## Restoration of electrophysiological behaviour and practical use for detailed network modelling

As demonstrated in Fig 7, cutting off the dendritic arbour of neurons is likely to lead to hyperexcitability in the electrophysiological model, even though the distribution of ion channels is similar in both the cut and the original neuron. The variability in firing behaviour of neurons with similar ion channel distributions has long been recognised [6]. Neurons that differ only in the geometry of their dendritic arbours produce a wide range of different spiking patterns. Typically, the smaller the neuron, the higher the spiking frequency. Therefore large neurons tend to be less excitable due to their lower input resistance/higher input conductance [67]. This is exactly what happens in Fig 7, as cutting away some of the dendritic material results in a smaller dendritic arbour, which now has a higher input resistance, thus inducing hyperexcitability.

We were able to show that neuronal repair also restores the more detailed electrophysiological properties of a neuron (Fig 8). The more complex model of [65, 66] includes, for example, HCN channels that produce a sag current. The sag current and spike properties in the repaired neuron are much closer to the reference than in the severed version. Such electrophysiological properties have never been tested for any neuron repair algorithm.

To compensate for reduced excitability, large neurons receive more synaptic input (see [67]), whereas small neurons have fewer synapses, reducing the effective current received by the neuron. As we have shown in Fig 9, activation of synaptic inputs close to the soma (proximal inputs) leads to large variability in responses, whereas inputs to distal parts of the dendrite

appear to produce more consistent dendritic spikes. Therefore, repairing incomplete morphologies and thus restoring distal synaptic input sites may make the electrophysiological behaviour of dendrites more consistent. More importantly, we were able to reproduce the findings of [14], who showed that the synaptic threshold for NMDA spikes is higher in human pyramidal cells than in mouse pyramidal cells. However, in the case of distal synaptic inputs, the reduced NMDA spike threshold was not present in incomplete human pyramidal cell morphologies. However, when incomplete human dendrites were completed using our repair method (Fig 9C), the higher NMDA spike threshold (for distal synapses) was restored. Like Testa-Silva and colleagues [14], we also found that the differences in the NMDA spike threshold were related to differences in dendritic diameter, which is increased in humans compared to mice. The addition of artificial dendritic material by the repair algorithm increases the average diameter in the distal dendrites of the human extended morphology (green) compared to the incomplete human neuronal morphology (black) (*c.f.* Fig 9D). The larger diameters explain the higher threshold for NMDA spike generation in repaired human dendrites than in incomplete human and in mouse dendrites.

In terms of dendritic geometry, not only differences in dendritic length, but also changes in topology such as the branching pattern significantly affect the firing behaviour of a neuron [9]. Dendritic topology also seems to have an effect on the type of firing, which can be expressed as bursts or regular spike trains [7]. The study by **Ref**. [9] suggests that changes in the dendritic geometry and topology, which are common in Alzheimer's disease, epilepsy and people with intellectual disabilities, have a significant impact on firing behaviour and therefore on information processing and cognitive ability. Therefore, restoring the missing parts of incomplete dendritic morphologies with our repair tool can restore the original firing behaviour of a neuron. The algorithm can be applied to the incomplete morphologies that are available in the databases of the Blue Brain Project [91] and the Allen Brain Atlas Data Portal (https://portal.brain-map.org). Combined with robust and generalisable biophysical models [60, 67], such improved morphologies could be used for large-scale network modelling.

## Limitations of our model and possible extensions

The repair of our software tool is based on the distribution of target points within a selected volume. These points are then successively connected to the existing reconstruction based on wiring optimisation constraints of cable length and conduction speed [41]. The volume can be chosen arbitrarily by the user. While this approach is highly flexible and gives the user complete freedom to choose where to grow the morphology, it does place an emphasis on the user's experience, anatomical knowledge and intuition. As shown in S2 Fig, the user needs to have in-depth knowledge of where to grow the missing dendritic material. In all cases, repaired dendrites should be considered as a model prediction that is useful for improving incomplete reconstructions but requires further experimental testing. The possibility to use a suitable reference image helps to assess where the boundaries of an intact morphology are and where certain parts are missing. A reference facilitates the repair of neurons from brain regions such as CA1 in the hippocampus where the sizes, shapes and anatomical layers are well defined, giving the user a clear indication of where somata are located and where to grow dendrites (as shown in the CA1 region; Fig 6). Less well-laminated and defined regions may be more challenging for the context-based neuronal repair. To overcome this problem, data-driven predictions for species-, cell-type- and region-specific anatomical boundaries based on morphological statistics would also need to be implemented. Such an algorithm would rely on a database containing reconstructions of many morphologies of different neuron types, regions and species. To predict the most likely complete boundary of a given input neuron, its type and region of

origin would have to be specified by the user. Based on the database, an average boundary could be calculated and scaled to the size and dimensions of the input neuron.

Our algorithm is unlikely to be suitable for repairing astrocytes or other glial cells, as it has not been validated for growing them specifically. It is currently unclear whether glial cells follow similar growth rules to neurons, but a recent study published a 3D editing tool for glial cells [92] to facilitate future detailed simulations of glial cells. If future research can confirm that MST_tree is indeed suitable for glial cell repair, the fix_tree function would need to be re-evaluated in this regard. Alternative growth algorithms may be incorporated into the fix_tree function in the future.

As the algorithm uses the uploaded incomplete morphology to automatically determine growth parameters such as the balancing factor, vastly incomplete morphologies can lead to inaccuracies. A morphology with very little dendritic material left is a challenge when trying to estimate growth parameters. Importantly, repaired morphologies can only be used to make predictions. It is important to realise that when one completes a dendritic tree based on the statistics of the remaining tree, homogeneous statistics throughout are assumed.

It is important to note that the repairs made by our algorithm are not perfect. Visually, they do not always resemble their biological counterparts exactly, as can be seen in Figs 2A and 2B and 3E. These examples are morphologies with low *bf*. In terms of the morphological statistics the algorithm was not able to perfectly replicate the Sholl profile of the reference neuron (Fig 5A). This mismatch was, at least in part, due to the stochastic nature of the algorithm. Nevertheless, the match was close, but in the most distal parts of the dendrites, the number of branches was slightly too high. In addition, the volume occupied by the repaired dendrites was slightly smaller compared to the reference morphology. Due to these discrepancies we find that we were able to restore the electrophysiological behaviour well, but the gain in the F-I curve of the repair did not match the reference exactly (Fig 7A–7C *inlays*). As for the more detailed electrophysiological properties the repair also represents a close match to the reference (Fig 8). The growth algorithms can be further refined in the future, e.g. based on developmental data [42, 43].

## Relationship to other morphological models

While there have been experimental studies investigating how *in vivo* neurons respond to injury and subsequently regrow and repair the damaged dendrites [52–54], artificial repair tools such as NeuroMorphoVis [36] mostly focus on removing artefacts that occur during the reconstruction process. Such artefacts include abrupt changes in dendritic thickness at bifurcations, soma profile adjustments, crossing neurites, and dendrites that are disconnected from the soma. Previous publications [37, 93] have presented a neuron repair tool in *NeuroR*. Their algorithm focuses on growing from severed ends only and has been validated on a single layer 2 pyramidal neuron. Importantly, such repair algorithms should restore the dendritic morphology as well as the original electrophysiological behaviour of a neuron in order to improve scientific inferences about neuronal functionality based on the repaired data. To date, there has been no electrophysiological validation of morphological repair algorithms. Our approach is therefore unique in that it uses principles of wiring optimisation, is generalisable and can be easily applied to any neuron type or species, and is capable of extending the dendritic arbour to create entirely new artificial sections. The easy-to-use graphical user interface allows the repair of incomplete or otherwise unusable morphologies. This also improves electrophysiological behaviour of reconstructed morphologies.

Morphological computational models mostly describe the growth as a stochastic process that depends on the branching probability, the number of branching events and the number of

segments [38–40]. It has recently been shown that a sequential stochastic growth and retraction algorithm is able to generate dendritic trees of *Drosophila* larval sensory neurons that are realistic in terms of both function and optimal wiring [42, 43], see also [94]. Similarly, building on the *TREES toolbox* [46], our repair tool also takes wiring optimisation into account. Therefore, switching between different neuron types with different wiring constraints can be done by adjusting a single free parameter, the balancing factor *bf*, which determines the neuron type specific optimal balance between cable length and conduction speed (Fig 2A). Using a limited set of parameters is the best way to implement a model if one wants to avoid overfitting problems [5]. Such simplicity makes our tool adaptable and easy to generalise to different morphologies and helps to understand whether certain neuron types optimise their dendrites primarily for material or conduction costs. However, our tool does not take into account the interactions between different neurons during growth, as do other morphological models such as CX3D [95, 96] and the one in the reference [97]. Here the authors [97] used an activity-driven algorithm where neuronal growth was determined by the activity of nearby potential synapses. The approach of CX3D focused on chemical gradients and mechanical forces that can generate layer-specific branching patterns. A similar morphological model, NETMORPH, in **Ref**. [98], growed neurons based on a stochastic branching outgrowth mechanism that does not use any extracellular cues. Modelling and completing multiple neuron types is likely to be more difficult using these alternative approaches, as the neuronal branching patterns in these models depend on many parameters.

## Conclusion

The *TREES toolbox*, extended by the new `fix_tree` function, allows for a range of investigations of dendritic anatomy, both during growth and repair, using synthetically grown dendritic structures. The morphological, and by extension functional, changes following cut and repair have not been extensively studied *in vivo*, and can be addressed *in silico* using our repair tool for both synthetic cell models and biological reconstructions. By making our tool widely available to the scientific community, datasets of human neuronal reconstructions could be improved and expanded. Such datasets could eventually provide the insight we need to understand what makes the human brain different from other species.

## Materials and methods

### Regrowth of lesioned class IV da-neurons of *Drosophila melanogaster*

We reconstructed the lesion paradigm, regrew the missing branches to re-cover the target area of the cell, and assessed the differences in morphology using statistical parameters. To study the bimodal distribution of regeneration from the lesioned stem and invasion we severed random dendritic subtrees of *Drosophila* da neurons, Purkinje cells and granule cells with lengths between $50\mu m < L < 1,000\mu m$ for 500 trials. Using the repair tool we regrew these 500 morphologies based on the volume previously occupied by the cut branches. To avoid a bias toward regeneration or invasion, target points were distributed within the growth volume with a given margin of $R_d$ away from any point of the lesioned neuron. To assess the distribution of regrowth, we determined what percentage of the regrown dendritic material was regenerated from the lesioned stem. The different growth modes of the GUI, and in particular the `fix_tree` function that is at the heart of the repair tool, are described in more detail in the next section.

## The `fix_tree` function of the repair algorithm

Based on the regrowth algorithm for *Drosophila* neurons (see above), we developed a stochastic model of regrowth after dendritic lesions in mouse and human CA1 pyramidal neurons using custom code implemented in the MATLAB-based *TREES toolbox* [46]. The `fix_tree` algorithm and UI are, like the *TREES toolbox*, only available in MATLAB (`fix_tree` and `fix_tree_UI` require the `inpolyhedron` function in MATLAB in order to operate. This was also explicitly mentioned in the YouTube tutorial). The supported file type is the ".swc" and ".mtr" format of the *TREES toolbox*, which ".asc" files can be converted to using the `neurolucida_tree` function of the *TREES toolbox*.

The repair algorithm is based on the minimum spanning tree (MST) function (`MST_tree`) from the *TREES toolbox* [41]. A tree is the representation of the morphology of a neuron by a set of nodes and an adjacency matrix defining the connections between these nodes. The distance between two consecutive nodes was adjusted by resampling the tree to achieve a distance between neighbouring nodes of $1\mu$m without significantly changing the branching morphology. The missing dendrites were regrown by distributing the target points over an area/volume $V$, which is an input to the function. To match the clustering of branch and termination points in the input neuron, the density profile of its spanning field is analysed and random clustered points are distributed accordingly using a Monte Carlo approach (available in the *TREES toolbox*). The number of target points *Npts* required is estimated by evaluating the density of branch points in the input neuron along with the size of the area/volume $V$. `MST_tree` then connects these points successively to the existing input neuron using a cost function [41] that depends on the balancing factor *bf*, which weights the conduction time (path length cost) against the material cost (wiring cost).

$$total\ cost = wiring\ cost + bf \cdot path\ length\ cost$$

The balancing factor *bf* is estimated by analysing the original input morphology using the *bf_tree* function in the *TREES toolbox* [59]. The maximum distance a single connection can span is limited by the growth threshold $G_{thr}$, which is calculated by measuring the part of a straight line $m$, passing through the neuron root $R$ (soma) and the point lying between the mean volume coordinate $V_{mean}$ and the volume coordinate furthest away from the root node $V_{far}$, that lies within $V$.

$$Q = mean(V_{mean}, V_{far})$$
$$m = \{\vec{x} = \overrightarrow{OR} + t \cdot \overrightarrow{RQ} \mid \vec{x} \in V\}$$

The values of $t$ must be chosen so that $m$ lies within $V$. New dendrites can grow from any point in the input tree within the range of $G_{thr}$ (biological growth). Biological growth is the first of two growth modes available in the `fix_tree` function, which fills the space by growing from lesioned or intact parts of the dendritic arbour. Alternatively, the algorithm can grow new dendrites exclusively from incomplete terminals of the neuron's branches (conserved growth), repairing a missing part of a severed neuron. Such incomplete terminals must be specified by the user with their exact coordinates in the uploaded morphology file. The algorithm only selects incomplete terminals for growth that are in close proximity to the growth area/volume $V$. The maximum distance an incomplete end can have to $V$ depends on the size of the original input tree. Additionally, the noninvasive/conserved growth mode has an option (main growth) specifically designed for severed apical dendrites of pyramidal neurons, since they usually feature one or more prominent main apical dendrites [26, 31]. These grow approximately in a straight line from the root of the tree. If main growth is enabled, the

algorithm will determine the thickest incomplete terminals in relation to all incomplete terminals and grow a main branch from these first, up to approximately 95% of the length of the growth volume. The direction and distance the main branch will grow is estimated by the same straight line *m* that was calculated earlier. *m* serves as a template for the main apical branch. The algorithm then proceeds as before, allowing dendrites to branch from the newly added main apical section.

In addition to the input neuron to be repaired, a reference morphology (if available) can be passed to the function. The algorithm then matches the number of branch points *NBr* of the repaired neuron to *NBr* in the reference neuron or to an arbitrary number (greater than *NBr* in the input neuron) by iterating over the growth process but successively adding more target points until the desired number is reached. The `fix_tree` function analyses the input neuron to set the growth parameters. It also distributes the target points in the growth volume, and identifies the incomplete (cut) terminals of the input neuron to restrict growth to those locations if needed. `fix_tree` then calls the `MST_tree` function, iterating with different numbers of target points until the number of branch points matches the desired value as closely as possible. `fix_tree` then edits the output tree, applying a jitter, adjusting diameters, and more.

The area/volume *V* for the repair dendrites to grow into is an input to the function and can be any set of user-defined 2D or 3D points. The volume is then defined by using the boundary function in *MATLAB*, which uses $\alpha$-shapes [99] to determine the outline of a set of points. How tightly the boundary fits is determined by a single parameter $\alpha$, where $\alpha = 0$ is the convex hull and $\alpha = 1$ is the tightest boundary.

To better match the appearance of the existing input neuron, low-pass filtered spatial noise is imposed on the coordinates of the grown dendrite as a spatial jitter. To achieve realistic diameter values for the grown dendrites, a quadratic taper is applied using the quadratic tapering algorithm of the *TREES toolbox* developed in **Ref**. [100]. The taper parameters are estimated based on the original existing morphology reconstructions. The repaired morphology is then tapered using these estimated parameters scaling down towards a minimum diameter in the terminal branches of the morphology as proposed in **Ref**. [101]. Since towards the very tips of the dendrites the diameters level off to a constant value, depending on the species, any diameters that fall below an adjustable threshold are set to that threshold value. Optionally, the morphology can be pruned to a desired dendritic length (*e.g.* length of a reference morphology) by first matching *NBr* and then trimming any excess material. By default, all parameters are estimated by analysing the morphology of the input neuron. The main parameter of the `MST_tree` function, the balancing factor *bf*, is estimated by analysing the root angle distribution as introduced in **Ref**. [59].

The GUI `fix_tree_UI`, for easy access to the `fix_tree` function, was programmed in the GUIDE MATLAB environment with a custom design interface (see Fig 4). The GUI can be accessed by running `fix_tree_UI` which uses the `fix_tree` function.

## Electrophysiology (*T2N*)

For electrophysiological compartmental modelling we used the previously developed *T2N* (*TREES*-to-*NEURON*) software interface [60] in MATLAB which links the compartmental modelling package *NEURON* [102] and the *TREES toolbox*. *T2N* allows for the creation and use of existing complex electrophysiology models, many of which are readily available from https://senselab.med.yale.edu/modeldb [103]. Any morphology in the *TREES toolbox* can be uploaded to *T2N* and is then equipped with ion channel conductances specified by the biophysical model. We simulated somatic current injections with a duration of 500ms and

ramping intensity for both mouse and human morphologies. Current clamps were performed on the reference, the repaired and the artificially cut morphologies respectively in order to compare their behaviour. We used a biophysical model from **Ref**. [64], previously imported into *T2N*. The model in **Ref**. [64] incorporates four active voltage channels (conductances). These channels include the following: a voltage-gated $Na^+$ channel, a delayed rectifier $K^+$ channel, a distal A-type $K^+$ channel with an elevated half-inactivation voltage and a proximal A-type $K^+$ channel. The model distributes these ion channels along the dendrites as a function of the length of the direct path to the soma. As the repair extended the cut dendrites, we stretched the ion channel distributions along the newly formed dendrites accordingly. In additional simulations (Fig 8), we have used an active model from **Refs**. [65, 66] that contains a set of active channels which are described in more detail in the next paragraph. The model in **Ref**. [64] includes a weak excitability version that follows a uniform distribution, which was used to model the delayed rectifier $K^+$ and the $Na^+$ channels. Following the experimentally reported sixfold increase in conductance along the apical dendrites, the A-type $K^+$ current was modelled accordingly. The result is linearly increasing slopes of variable nature between soma and tuft for different morphologies. The regions of the apical dendrites were defined as follows: the boundaries for the apical trunk (proximal apical) were set to contain 3.14% of the total apical length. The medial apical dendrites contain 36.27%, the distal 68.90% and the tuft 100% of the total apical length. The dendrites were divided at path distances of approximately $100\mu m$, $300\mu m$ and $500\mu m$.

The same current clamp procedure with different stimulation current increments was applied to a mouse CA1 pyramidal neuron (data by **Ref**. [68]) using a different compartmental model in **Refs**. [65, 66]. The model by Poirazi et al. is more detailed than the one by Jarsky and includes HCN and slow $K^+$-channels that allow for the simulation of more intricate electrophysiological properties such as the sag current. The model by Poirazi et al. incorporates 6 different $Ca^{2+}$, 5 $K^+$, 1 $Na^+$, 1 h-current and 2 Hodgkin-Huxley-channels. The neuron morphology was equipped with an artificial soma using the *soma_tree* function of the TREES toolbox and an artificial axon. The soma dimensions were chosen such that the surface area equaled approximately $\sim 560\mu m^2$. The axon in the form of a straight line had a length of $630\mu m$ with six $100\mu m$ myelin segments. The axon featured a hillock, an initial segment and 5 nodes of Ranvier with an average axon diameter of $0.5\mu m$. The hillock started at $2\mu m$ in diameter, tapered down to $0.5\mu m$ and was $10\mu m$ long. The initial segment had a length of $15\mu m$ whereas the nodes of Ranvier were $1\mu m$ in length. To analyse the sag current, small negative current stimulations were performed. The electrophysiological properties were extracted from the voltage traces using the *findpeaks* function in MATLAB.

To simulate synaptic dendritic spikes, we implemented AMPA and NMDA synapses at different locations on the basal dendrites of three morphologies (mouse, human and human extended). The simulations were again carried out using the model in **Ref**. [64], but with all active ion channel conductances switched off, leaving only the passive properties of the model. The dendritic diameters on these morphologies were adjusted to eliminate any artefacts that arose during the reconstruction process when using Neurolucida 360 (MBF Bioscience). Synaptic stimulation was carried out at different Euclidean distances from the soma based on the maximum possible Euclidean distance from the soma of the basal dendrite. The distances were thus scaled for the different morphologies respectively, since the human morphologies are much larger than mouse morphologies. The procedure is designed to expand on what was previously done by Testa-Silva et al. [14], who measured NMDA spikes in human and mouse layer 2/3 pyramidal neurons at only one fixed distance ($150\mu m$ from the soma) for mouse and human. To account for morphological variability, 10 different sites were simulated for each distance and the average was calculated. The stimulation strength was determined by the

number of synapses, which were distributed over segments of 20$\mu$m in length. We then recorded the dendritic spike response for different numbers of synapses with $gAMPA$ = 25pS and $gNMDA$ = 500pS at the stimulation site. We also measured the diameters for each part of the 20$\mu$m sections.

## In brief

We use morphological modelling inspired by the regeneration of various artificially cut neuron types and repair incomplete human and nonhuman neuronal dendritic reconstructions.

## Highlights

- Optimal wiring-based growth algorithm replicates regrowth of artificially cut dendrites

- The growth algorithm repairs cut dendrites in incomplete reconstructions

- The algorithm works for diverse neuron types in multiple species

- The repair of morphology restores original electrophysiology

- The repair of morphology supports simulations of high synaptic thresholds for NMDA spikes in human dendrites

- The repair tool with user interface is available in the *TREES toolbox*

## Supporting information

**S1 Fig. Hippocampal CA1 region of the mouse. A,B**, Confocal microscope image of the mouse hippocampus with stained pyramidal neuron morphologies, region of interest (ROI) and marked layers (SLM: stratum lacunosum moleculare, SR: stratum radiatum, SP: stratum pyramidale, SO: stratum oriens). **C**, Morphology reconstruction overlays with marked layers. **D**, Magnified ROI with marked layers. **E**, Magnified ROI with marked layers and example of reconstructed morphology overlay. Imaging data were taken from [26]. Scale bar = 75 $\mu$m in **D, E**. Scale bar = 230 $\mu$m in **A, B, C**.
(TIF)

**S2 Fig. Hippocampal CA1 region of the human with marked growth volumes. A**, Image showing 3D human CA1 pyramidal neuron reconstructions in **Ref**. [26]. The shaded green areas are example regions where Benavides-Piccione et al. [26] knew dendrites should be present during the reconstruction process but they could not reconstruct them since the dendrites were not visible in the confocal microscope images. The green regions were used as an inspiration when extending the human neuron reconstructions in Fig 6E. Scale bar = 145 $\mu$m.
(TIF)

**S3 Fig. The effect of different cuts in the apical dendrite on neural firing behaviour. A**, F-I curve for reference and repaired mouse CA1 pyramidal neurons. 20 different lesions (magenta line) were performed on the reference neuron (black line) and then repaired (green line). The

magenta line shows the average of the 20 different cuts with the standard deviation as the shaded area. The repairs are shown in green with the standard deviation as the shaded area. See Fig 7A which shows the procedure for one specific cut and repair. **B**, Same as in **A** but for a human neuron morphology (see Fig 7B).
(TIF)

**S4 Fig. NMDA spikes in human neurons with unscaled synapse distance from the soma. A**, Peak NMDA spike voltage for a mouse (blue), human (black) and human extended (green) morphology. The peak NMDA spike voltage is measured for different numbers of synapses at different distances from the soma in the basal dendrite, given as a percentage of the maximum possible distance in the basal tree. Unlike in Fig 9 however, the distance from the soma for the human extended neuron (green) was exactly equal to that of the human (black) neuron (see text in **A**, the absolute stimulation distance for the two human neurons is given in $\mu m$). For each distance 10 different locations at that distance were tested (transparent dashed coloured lines). The average is shown as a solid line. The synapses were distributed over $20\mu m$ sections (same procedure as in Fig 9). **B**, Dendritic diameters for the locations described in **A**, with mean and standard deviation.
(TIF)

## Acknowledgments

We are grateful to Y. Song for communications in the initial phase of the project and to Martin Mittag for his support in NMDA spike simulations.

## Author Contributions

**Conceptualization:** Moritz Groden, Hannah M. Moessinger, Barbara Schaffran, Hermann Cuntz, Peter Jedlicka.

**Data curation:** Moritz Groden, Ruth Benavides-Piccione, Hermann Cuntz.

**Formal analysis:** Moritz Groden, Javier DeFelipe, Ruth Benavides-Piccione, Hermann Cuntz, Peter Jedlicka.

**Funding acquisition:** Hermann Cuntz, Peter Jedlicka.

**Investigation:** Moritz Groden, Hannah M. Moessinger, Barbara Schaffran.

**Methodology:** Moritz Groden, Hermann Cuntz.

**Project administration:** Moritz Groden, Hermann Cuntz, Peter Jedlicka.

**Resources:** Javier DeFelipe, Ruth Benavides-Piccione, Hermann Cuntz, Peter Jedlicka.

**Software:** Moritz Groden, Hermann Cuntz.

**Supervision:** Javier DeFelipe, Ruth Benavides-Piccione, Hermann Cuntz, Peter Jedlicka.

**Validation:** Moritz Groden, Hermann Cuntz, Peter Jedlicka.

**Visualization:** Moritz Groden, Ruth Benavides-Piccione, Hermann Cuntz.

**Writing – original draft:** Moritz Groden, Hannah M. Moessinger, Barbara Schaffran.

**Writing – review & editing:** Moritz Groden, Javier DeFelipe, Ruth Benavides-Piccione, Hermann Cuntz, Peter Jedlicka.

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
