## [Decision Letter · Decision Letter 0]

7 Aug 2023

Dear Dr. Groden,

Thank you very much for submitting your manuscript "A biologically inspired repair mechanism for neuronal reconstructions with a focus on human dendrites" for consideration at PLOS Computational Biology.

As with all papers reviewed by the journal, your manuscript was reviewed by members of the editorial board and by several independent reviewers. In light of the reviews (below this email), we would like to invite the resubmission of a significantly-revised version that takes into account the reviewers' comments.

In particular, a more rigorous approach is needed to back up the claims that the neurons have been adequately repaired. The reviewers suggest several metrics that could be used to make a more compelling case.

We cannot make any decision about publication until we have seen the revised manuscript and your response to the reviewers' comments. Your revised manuscript is also likely to be sent to reviewers for further evaluation.

Sincerely,

Abigail Morrison

Academic Editor

PLOS Computational Biology

Daniele Marinazzo

Section Editor

PLOS Computational Biology

Reviewer's Responses to Questions

**Comments to the Authors:**

Reviewer #1: The authors test their morphology repair algorithm on human neurons and make a few predictions about the effects on electrophysiological properties. While the tool is useful and the aims are interesting, there is a general lack of sufficient quantitative details in the results, and some key analysis is missing.

Major:

1. Fig2 D-F: authors should clarify what is the corresponding experimental profile, to assess the algorithm performance

2. Authors should clarify the details of their algorithm visually and conceptually, and what bf means, and the implications of the distribution they plot.

3. Fig 3 should include metrics such as given in Fig2C to support claim of success. It’s not clear why authors focus only on % regrowth from cut branch.

4. Fig 5 is rather qualitative, and should include metrics to compare the original/cut/repaired morphologies and support the claim of success. e.g. the metrics shown in fig 6, and/or distance between Sholl profiles, number of peaks etc. For example, bottom left reconstruction has one large second peak vs 3 small peaks in the original.

5. Fig 5 top right, basal dendrites cut seem to be middle sections of the dendrites. Authors should clarify that case, and potentially the utility of their algorithm for such cuts.

6. The results text lacks supporting quantitative analysis and statistics, e.g. corresponding to Fig 3 and Fig 6 – what is the correlation or RMSE between cut/original vs repaired/original, what is the effect size / statistical significance? Authors should use that to help the reader assess the lesser success for basal length per segment, for example.

7. To support the success of the algorithm, in fig 6 Authors should plot performance on a larger dataset than just 6 morphologies, to support statistical testing and claims, and also on different cell types.

8. Fig 7 – should include quantitative aspects of the reconstruction. Even the qualitative nature of the reconstruction is unclear.

9. Fig 8 – should include quantitative aspects – firing rate of the ref/cut/repaired cases, FI curve/gain, and the corresponding text should include stats and significance values. Also for 8C – what is the predicted change in rate or FI curve/gain?

10. Fig 8 - authors should check the effect of cut and repair on other electrophysiological properties e.g. sag current, adaptation index.

11. Fig 8 should examine the effect of different cuts as the authors did in fig 2, and plot the resulting metrics.

12. Fig 9 – authors should account for the cutting of the mouse morphologies as well, for a proper comparison of repaired prediction between species. Also, authors should highlight what the prediction from C is.

The relevance of part D is unclear and should be either be used in the context of the NMDA analysis or removed from the figure.

13. Fig 9 - the repaired human morphology has a different % from soma location, how does that affect the analysis? Authors should analyze and plot results for absolute distance comparison as well, to address this confound.

14. Authors should illustrate and discuss the limitations of their algorithm. Are there cases where it struggles?

15. Authors should clarify the dendritic ion channels and their distributions, are they a function of the relative or absolute distance from soma, and how does the repair affect the distributions?

Minor:

1. Number of figures can be reduced by combining e.g. fig5+6 (with fewer than 6 examples if necessary)

2. Fig 2e – for better clarity, authors should instead plot length of cut on x axis and something like a violin plot of the distribution of % growth from cut

3. Fig 6 caption should clarify what measure they refer to by diameter per segment – is it average?

4. Fig 9 should use a different color for mouse, to avoid confusion with previous figures

5. Authors should clarify if users can use the tool in other languages such as python, and what file types are supported.

Reviewer #2: Manuscript Number: PCOMPBIOL-D-23-00929

Full Title: A biologically inspired repair mechanism for neuronal reconstructions with a focus on human dendrites

Reviewer comments:

The manuscript presents an algorithm provided in TREES Toolbox to repair incomplete dendritic reconstructions either by regenerating from the cut end of the branch or invading from adjacent branches. This algorithm and provided graphical user interface are highly needed in the field, especially related to human cells. The authors have used the algorithm with both mouse and human neuron morphologies. They have used the algorithm with whole neuron morphologies by cutting a part of the morphology away and then repairing it, and finally comparing the original (reference) morphology to the cut and repaired ones. They have also repaired morphologies that never were whole neuron morphologies. Electrophysiological behavior was also simulated with the reference, cut, and repaired morphologies, but more detailed analysis is needed to understand in detail how similar the membrane potentials are in these different cases. Figure 9 shows interesting results when comparing mouse, human, and human extended morphologies. After testing the fix_tree_UI a little bit, the algorithm is working but I could have received better results with better instructions. A video file and a manual showing how to run and use the tool would be helpful for the users.

I have provided below my comments in more detail.

1) Provide in the website of TREES Toolbox a video file and manual how to use fix_tree_UI for easy use of the tool. Provide information and figures in the manual how the target points should be chosen because it took long time to get realistic-looking dendrites. In addition, on l. 88: Please, be more specific by explaining what you mean by "target points".

2) Please, test the tool for astrocytes and possibly also for other glial cell types. At least discuss about it in discussion and if the tool is succeeding with glial cells, please add a figure about the morphology regeneration and invasion, testing the functionality is not needed.

3) Consider discussing how your tool is different to other tools that are not cited in the manuscript, such as NETMORPH and CX3D. Furthermore, consider adding references related to dendritic growth models, such as Kirchner et al. (Dendritic growth and synaptic organization from activity-independent cues and local activity-dependent plasticity).

4) In many occasions, you use "cell type" when actually "neuron type" would be more appropriate. Consider changing this to sentences that are not specific enough. You often write "human morphology" and "mouse morphology", but please be more specific and use "human neuron morphology" and "mouse neuron morphology".

5) Please, add Data availability statement.

6) Figure text of Figure 2: Explain more clearly in A and B, Left, that they show the branches in magenta that will be severed: e.g., "Reference Drosophila larva Class IV morphology in which the branches that will be severed deliberately are marked in magenta." Write "B, Right, Sample" instead of "B, Sample".

7) Figure 3: It seems that the bars are not adding up to 500, so please make sure the y-axis is correct.

8) Figure text of Figure 3: Explain in Methods clearly and also shortly in Results or in this figure text what is the difference between fix_tree and MST_tree. Explain here if one or both are used in A-C.

9) Figure text of Figure 4: Please, be more specific where items 1-2 showing on top of the figure are done because those are not provided in the GUI. Specify also that extra tools (Inpolyhedron) are needed for this. Write "Graphical-User-Interface" as "Graphical User Interface".

10) Figure text of Figure 6: To make the figure more clear, please explain the abbreviations used in the titles of the subfigures. Thus write open the titles as you have done in the results section: Total number of branch points, total dendritic length, dendritic length per segment and the diameter per segment for apical and basal arbours.

11) Figure 8: Please add more analysis of the membrane potentials so that the reader can see in detail how similar the membrane potentials are in reference, cut, and repaired morphologies: e.g., ISI, frequency of spikes, peak value, etc. Only after these results, the possible similarity can be stated.

12) Figure text of Figure 9: Explain in the figure text, if or if not the human morphology presented is the same human morphology that is extended. Add a comma before "respectively". "µm" should not be in cursive. Is "purple" actually "magenta"?

13) You could cite your figures more in Discussion.

14) The order of references in the sentences is not always following any rules, e.g. oldest articles first/alphabetical order. Please, note that PLOS Comput Biol uses numeric citation style.

15) l. 130-136: This paragraph could be reworded in such a way that the readers understand that the tool is working properly. The first sentence can give an impression that the invasion was undesirable. Furthermore, explain better that Fig. 2A, Left and 2B, Left show the original reference morphology where branches that will be severed are shown in magenta.

16) l. 160: You have explained the balancing factor (bf) also on l. 152-154. Maybe it would be more clear to define it well the first time you use it in Results, thus in l. 152-154.

17) l. 193-194: Specify in more detail what you mean by "arbitrary morphologies". Are other cells than neurons also possible?

18) l. 426: Please, use "Intellectual disability" instead of "mental retardation".

19) l. 431: Add a reference to "Allen Brain Atlas Data Portal".

20) l. 561: Please be more specific that the user has to run fix_tree_UI in MATLAB to use the tool and not fix_tree.

Smaller typos:

21) There are some grammatical mistakes in the GUI in the repair panel in the descriptions of the top two items (e.g. engabled).

22) Sometimes you write "TREES Toolbox" and sometimes "TREES toolbox". Same for "arbor" and "arbour".

23) l. 121: Add comma after "Branches".

24) Figures 2C and 4, write "Rep." as "Repaired".

25) l. 149: "mm" should not be in cursive. l. 271-272: "nA" and "ms" should not be in cursive. l. 294, 500, 587, 599, 602, 605: "µm" should not be in cursive. l. 571: "ms" should not be in cursive. l. 604: "pS" should not be in cursive.

26) l. 177: Write "has was cut" as "was cut".

27) Figure text of Figure 7: Write "intracellularly injected pyramidal cells" as "stained pyramidal cells". Write "B" as "B,".

28) l. 302, 308, 317: Is "purple" actually "magenta"?

29) l. 335-336: Add a comma before and after: "i.e. path length to the root (Cuntz et al., 2010)".

30) l. 403: Write "al. (2021)" as "al., 2021".

31) l. 462: Write "such as Abdellah et al. (2018)" as "such as NeuroMorphoVis (Abdellah et al., 2018)".

32) Last paragraph on page 33: "µm" should not be in cursive.

33) l. 581: Write "channel" as "channels".

34) l. 588: Add empty space between paragraphs.

35) l. 594-595: Write "euclidean" as "Euclidean".

36) l. 603: Should "length" be "in length"?

37) Figure text of Figure S1: Make "E" bold and not as cursive.

Reviewer #3: In Groden et al., inspired by biological regrowth in severed Drosophila dendrites, the authors propose that such approach could be deployed to repair incompletely reconstructed dendritic morphologies. The authors claim that their repair algorithm successfully recovered the incomplete parts of the dendritic trees and therefore could be used to simulate and predict their electrophysiological responsiveness. The algorithm itself is not novel as has been previously used and published to generate morphologies but is used and suggested for the first time to repair incompletely reconstructed morphologies. As I have pointed out below in my detailed comments that some other algorithms do exist which are deployed by Blue Brain Project team in repairing dendritic morphologies, the text in the manuscript does not capture it as it and claim that their algorithm is first one to repair the missing dendrites. Whether the existing algorithms do good job or not can be debated but should be mentioned for what those algorithms are meant to do. Besides that, the authors should improve some statistical analysis of morphological features to make effectiveness of their repair algorithm more outstanding.

DETAILED COMMENTS:

1. Lack of objective measure to support the claim that the morphology has been repaired successfully. Although providing evidence for bimodal regrowth generated by the repair algorithm is interesting, i am not convinced if it is essential to retrieve the functional properties of the neurons. Therefore better statistical comparison must be performed to compare uncut and repaired morphologies. e.g. when i see Purkinje cells in Fig 3B, the repaired one looks very different from the original one, so what makes the authors to claim that the repair algorithm worked? In Fig 3A, all the dendrites are reaching towards the top whereas the repaired ones are terminating far from the boundary. I understand that it might be difficult to establish the ground truth for repairing morphologies but I am sure its possible to compare multiple morphological features with statistical tests to report some objective measurements. The authors have tried to present morphological measurements in Fig.5 and 6. But in my opinion these statistics are not sufficient to claim successful repair. I clearly see higher branchings in Sholl distribution of first two repaired morphologies as compared to the original ones in Fig 5. Surprisingly, I don’t see that in Fig 6, so I am not sure how to explain higher branch crossings in Sholl analysis for apical tuft.

2. the dendritic repair algorithm presented in Anwar et al 2009 does aim to add new branches to existing dendritic trees. How thoroughly they validated their algorithm can be debated but such a tool already existed. Please clarify this in the top paragraph (L70-80) on page 7. Same in Discussion “Relationship to other morphological models” L460-469.

3. Comparison is Fig 2C is not clear to me especially for Nr of branches and Total dendritic length. I would suggest running statistics (between Cut and Rep groups) and report p-values to see if those two classes are significantly different from each other or not.

4. It is very interesting and reasonable approach to test repair algorithms on reconstructions where complete data is available and extend the approach to the reconstructions where complete data is not available but research scientists must use caution when making claims using such approach and the readers should be warned of the limitations in clear words. Extending this approach to repair incompletely reconstructed human neurons and simulating those neurons to provide predictions is a bold claim. As shown by the authors, the electrophysiological properties of these neurons are usually tested by current clamp recordings often at soma or occasionally at proximal apical dendrites, so matching the responsiveness at soma or proximal dendrites does not ensure matching characteristics at distal dendritic levels. I understand that these approaches are commonly used in neuroscience community. Authors should mention all these limitations so that the claim of successful repair does not look inflated.

**Have the authors made all data and (if applicable) computational code underlying the findings in their manuscript fully available?**

Reviewer #1: Yes

Reviewer #2: Yes

Reviewer #3: Yes

PLOS authors have the option to publish the peer review history of their article (what does this mean?). If published, this will include your full peer review and any attached files.

Reviewer #1: No

Reviewer #2: No

Reviewer #3: No
---

## [Decision Letter · Decision Letter 1]

2 Feb 2024

Dear Dr. Groden,

We are pleased to inform you that your manuscript 'A biologically inspired repair mechanism for neuronal reconstructions with a focus on human dendrites' has been provisionally accepted for publication in PLOS Computational Biology.

Best regards,

Abigail Morrison

Academic Editor

PLOS Computational Biology

Daniele Marinazzo

Section Editor

PLOS Computational Biology

Reviewer's Responses to Questions

**Comments to the Authors:**

Reviewer #1: The authors have addressed all of my concerns. It is now clearer and the analysis is stronger and more quantitative.

Reviewer #2: The authors have addressed all my concerns. However, I notices some typos in the new version that should be fixed:

1) I am not sure if references using "Ref. [number]" is allowed in PLOS Comput Biol. Should the sentences be changed into passive and/or move the reference number in the end of the sentence so that it is not part of the sentence text?

2) There were several typos in the new text added into this version. Please double check your text. Some of the typos are given below as examples:

-Cite also the original study of CX3D: Zubler F, Douglas R. A framework for modeling the growth and development of neurons and networks. Frontiers in Computational Neuroscience. 3(25), 2009. doi: 10.3389/neuro.10.025.2009

-New text has units in italics.

-Caption of Figure 8: typo in "freuency".

-Line 737: typo in "intrciate"

-Line 738: "+" in not in the superscript.

-Line 746: typo in "anaylse".

-Caption of Figure S3: "human morphology" -> "human neuron morphology"

-Figure S4: "sclaed" -> "scaled", "neruon" ->"neuron"

Reviewer #3: The authors have addressed all my concerns in detail. I am sure this study will be very useful for making further progress in the field of computational neuroscience/neuroanatomy and will be of interest to broad readership. I have no further comments.

**Have the authors made all data and (if applicable) computational code underlying the findings in their manuscript fully available?**

Reviewer #1: None

Reviewer #2: Yes

Reviewer #3: Yes

PLOS authors have the option to publish the peer review history of their article (what does this mean?). If published, this will include your full peer review and any attached files.

Reviewer #1: No

Reviewer #2: No

Reviewer #3: No

---

## [Editor Report · Acceptance letter]

20 Feb 2024

PCOMPBIOL-D-23-00929R1 

A biologically inspired repair mechanism for neuronal reconstructions with a focus on human dendrites

Dear Dr Groden,

I am pleased to inform you that your manuscript has been formally accepted for publication in PLOS Computational Biology. Your manuscript is now with our production department and you will be notified of the publication date in due course.

With kind regards,

Bernadett Koltai
